# Caveolin-1-mediated sphingolipid oncometabolism underlies a metabolic vulnerability of prostate cancer

Jody Vykoukal[1,2,9], Johannes F. Fahrmann[1,9], Justin R. Gregg[3], Zhe Tang[4], Spyridon Basourakos[4], Ehsan Irajizad[5], Sanghee Park[4], Guang Yang[4], Chad J. Creighton [6,7], Alia Fleury [1], Jeffrey Mayo[1], Adriana Paulucci-Holthauzen[8], Jennifer B. Dennison[1], Eunice Murage[1], Christine B. Peterson [5], John W. Davis[3], Jeri Kim[4,10 ✉], Samir Hanash [1,2,10 ✉] & Timothy C. Thompson [4,10 ✉]

Plasma and tumor caveolin-1 (Cav-1) are linked with disease progression in prostate cancer. Here we report that metabolomic profiling of longitudinal plasmas from a prospective cohort of 491 active surveillance (AS) participants indicates prominent elevations in plasma sphingolipids in AS progressors that, together with plasma Cav-1, yield a prognostic signature for disease progression. Mechanistic studies of the underlying tumor supportive oncometabolism reveal coordinated activities through which Cav-1 enables rewiring of cancer cell lipid metabolism towards a program of 1) exogenous sphingolipid scavenging independent of cholesterol, 2) increased cancer cell catabolism of sphingomyelins to ceramide derivatives and 3) altered ceramide metabolism that results in increased glycosphingolipid synthesis and efflux of Cav-1-sphingolipid particles containing mitochondrial proteins and lipids. We also demonstrate, using a prostate cancer syngeneic RM-9 mouse model and established cell lines, that this Cav-1-sphingolipid program evidences a metabolic vulnerability that is targetable to induce lethal mitophagy as an anti-tumor therapy.

[1] Department of Clinical Cancer Prevention, The University of Texas MD Anderson Cancer Center, 1515 Holcombe Boulevard, Houston, TX 77030, USA. [2] McCombs Institute for the Early Detection and Treatment of Cancer, The University of Texas MD Anderson Cancer Center, 1515 Holcombe Boulevard, Houston, TX 77030, USA. [3] Department of Urology, The University of Texas MD Anderson Cancer Center, 1515 Holcombe Boulevard, Houston, TX 77030, USA. [4] Department of Genitourinary Medical Oncology, The University of Texas MD Anderson Cancer Center, 1515 Holcombe Boulevard, Houston, TX 77030, USA. [5] Department of Biostatistics, The University of Texas MD Anderson Cancer Center, 1515 Holcombe Boulevard, Houston, TX 77030, USA. [6] Department of Bioinformatics and Computational Biology, The University of Texas MD Anderson Cancer Center, 1515 Holcombe Boulevard, Houston, TX 77030, USA. [7] Dan L Duncan Comprehensive Cancer Center Division of Biostatistics, Baylor College of Medicine, One Baylor Plaza, Houston, TX 77030, USA. [8] Department of Genetics, The University of Texas MD Anderson Cancer Center, 1515 Holcombe Boulevard, Houston, TX 77030, USA. [9]These authors contributed equally: Jody Vykoukal, Johannes F. Fahrmann. [10]These authors jointly supervised this work: Jeri Kim, Samir Hanash, Timothy C. Thompson. ✉email: jeri.kim@merck.com; shanash@mdanderson.org; timthomp@mdanderson.org

Elevated serum Cav-1 levels have been associated with high-risk prostate cancer[1], castration-resistance[2], and biochemical recurrence after prostatectomy[3]. We have also previously demonstrated the increased plasma Cav-1 to be associated with early disease reclassification in individuals with prostate cancer that initially present with clinically localized disease[4]. Cav-1 is the eponymous protein component of caveolae: bulb-shaped, 50–100 nm invaginations of the plasma membrane that are enriched in glycosphingolipids and cholesterol. Cav-1 also functions in organizing membrane microdomain composition and in modulating transmembrane signal transduction. Increasing evidence indicates that Cav-1 functions as an essential lipid chaperone to facilitate cellular lipid trafficking and homeostasis, endocytosis and exocytosis, and mechanoprotection of cell membranes[5–7]. Cav-1 is known to transport molecules including insulin, chemokines, albumin, and low-density and high-density lipoproteins (LDL and HDL)[8]. Recently, Cav-1 containing extracellular vesicles in white adipose tissue were found to traffic intracellular exchange of protein and lipid between endothelial cells and adipocytes in response to system metabolic state[9].

In cancer, the role of Cav-1 is dynamic and context-dependent[10]. Cav-1 has been demonstrated to regulate and promote the activities of receptor tyrosine kinases, G-protein coupled receptors, integrins and cadherins[11]. Expression of Cav-1 has been closely associated with aggressive phenotypes in various tumor types[12,13] and has been linked to epithelial–mesenchymal plasticity[14–17], tumor invasion and metastatic potential[18,19], and radiodrug and multidrug resistance[5,20].

Although Cav-1 has been associated with altered metabolism in prostate cancer[21–23], the mechanism by which Cav-1 effects metabolic rewiring has not been previously determined. In this study, we interrogate Cav-1 function in the context of prostate tumor metabolism and uncover an integrated metabolic program of enhanced lipid scavenging and differential ceramide metabolism active in prostate tumors that exhibit Gleason grade progression following initial enrollment into active surveillance. Importantly, this metabolic phenotype yields biomarkers of disease progression and identifies points of therapeutic susceptibility. Key features of this tumor supportive metabolic program include Cav-1 mediated lipid uptake, increased tumor catabolism of extracellular sphingomyelins, and altered ceramide metabolism coupled with increased glycosphingolipid synthesis and efflux of circulating Cav-1-sphingolipid particles that comprise cargoes indicative of intersection with mitochondrial remodeling. On the basis of these mechanistic findings, we test for potential actionable metabolic vulnerabilities by targeting Cav-1-mediated lipid scavenging and metabolism in a mouse model of aggressive prostate cancer.

## Results

### Plasma lipid signature predicts AS Gleason grade progression.

We conducted untargeted metabolomics analyses on clinically matched baseline plasma samples ($n = 16$ per group) prospectively collected from patients with clinically low-risk early stage prostate cancer undergoing AS who exhibited early disease progression (DP) (defined as upgrading of Gleason score (GS) and/or increased tumor volume on surveillance biopsy within 18 months after start of AS) or indolent disease (no progression for five or more years after start of AS). A total of 269 unique annotated metabolite features were identified in baseline plasmas from our discovery cohort; 14 features exhibited statistically significant (unadjusted Wilcoxon-rank sum test $p$-value < 0.05) receiver operating characteristic (ROC) area under the curve (AUC) values >0.7. Seven of the 14 features were complex lipids, in particular, sphingomyelins and glycosphingolipids (Supplementary Table 1). Although none of the

14 features remained individually significant after adjusting for multiple hypothesis testing, the significantly elevated lipids are biochemically linked to ceramide metabolism, suggesting a coordinated signal. Further, sphingomyelins and glycosphingolipids, in general, were elevated in baseline plasma samples of progressor cases (early DP) as compared to controls (no disease progression for a minimum of 5 years after start of AS) (Fig. 1a). Importantly, the detected sphingomyelins (SMs) and glycosphingolipids remained elevated in cases at DP as compared to follow-up time-matched controls (Fig. 1a). Intracase comparisons of sphingomyelins and glycosphingolipids observed at baseline vs. 12 months post-baseline were not statistically significant (Supplementary Fig. 1). We hypothesized that—given the known lipid transport functions of Cav-1[6]—the observed elevated plasma sphingolipid signature might be biologically linked to our previous findings of elevated plasma Cav-1 in the context of disease progression. To explore this, we expanded our analyses and performed untargeted metabolomics profiling on 459 baseline plasma samples prospectively collected from patients with early-stage prostate cancer undergoing AS. Consistent with our findings in the initial discovery cohort, multiple SMs and glycosphingolipids were positively associated (Hazard ratio >1.5) with DP based on GS (Fig. 1b; Supplementary Table 2). Next, we focused on those sphingolipids that exhibited statistically significant ($p < 0.05$) HRs and, using a logistic regression model, developed a signature panel comprising plasma Cav-1 and six sphingolipids that exhibited positive ß-estimates in the logistic regression model: SM(40:2), SM(44:2), lactosylceramide(32:0), lactosylceramide(36:0), trihexosylceramide(34:1), and hexosylceramide (40:0). Using log rank test statistics from the Cox model[24], we calculated an optimal cut-off point for the plasma Cav-1-sphingolipid signature that would yield the greatest difference between subjects on AS that exhibited disease progression (defined as upgrading of Gleason score and/or increased tumor volume) from those that did not. This resulted in a cut-off value of 4.33. We subsequently assessed the association of the signature with progression free survival using Cox-proportional hazard models. In a multivariate analyses, adjusted for age, 5-alpha reductase treatment, and baseline tumor volume, AS subjects with a plasma Cav-1-sphingolipid signature score >4.33 exhibited statistically significantly worse disease progression free survival as compared to those with a plasma Cav-1-sphingolipid signature score ≤4.33 (HR: 2.70, 95% CI: 1.75–4.16, $p$-value: <0.001) (Supplementary Table 3). Notably, non-proportionality hazard model tests yielded non-significant $p$-values. Of relevance, in our analyses BMI was not associated with increased risk of DP (HR: 1.02, 95% CI: 0.40–2.64, $p$-value: 0.965) (Supplementary Table 3) indicating that our lipid signature is unlikely to be biased by obesity. Kaplan–Meier survival curves depicting progression free survival for participants with plasma Cav-1-sphingolipid signature scores below (<4.33) or equal to or above the cutoff (≥4.33) are shown in Fig. 1b.

To better define the relationship between lipid metabolism and Cav-1 within the context of the Cav-1-sphingolipid signature, we first compared gene expression profiles reflective of lipid managing apparati and mRNA expression of CAV1 in 333 prostate tumors using The Cancer Genome Atlas (TCGA). High CAV1 mRNA expression was found to be positively associated with genes annotated to ontologies related to lipid scavenging and metabolism, glycosylceramide metabolic process as well as the ceramide pathway (Fig. 1c; Supplementary Data 1 and 2). We additionally explored whether elevated CAV1 and associated lipid managing apparti would associate with previously defined molecular subtypes (iClusters) of primary prostate cancers[25]. Our results indicated that high CAV1 mRNA expression was predominately associated with iCluster 3, which is characterized by elevated PI3K/AKT, MAP-Kinase, and receptor tyrosine kinase activity[25].

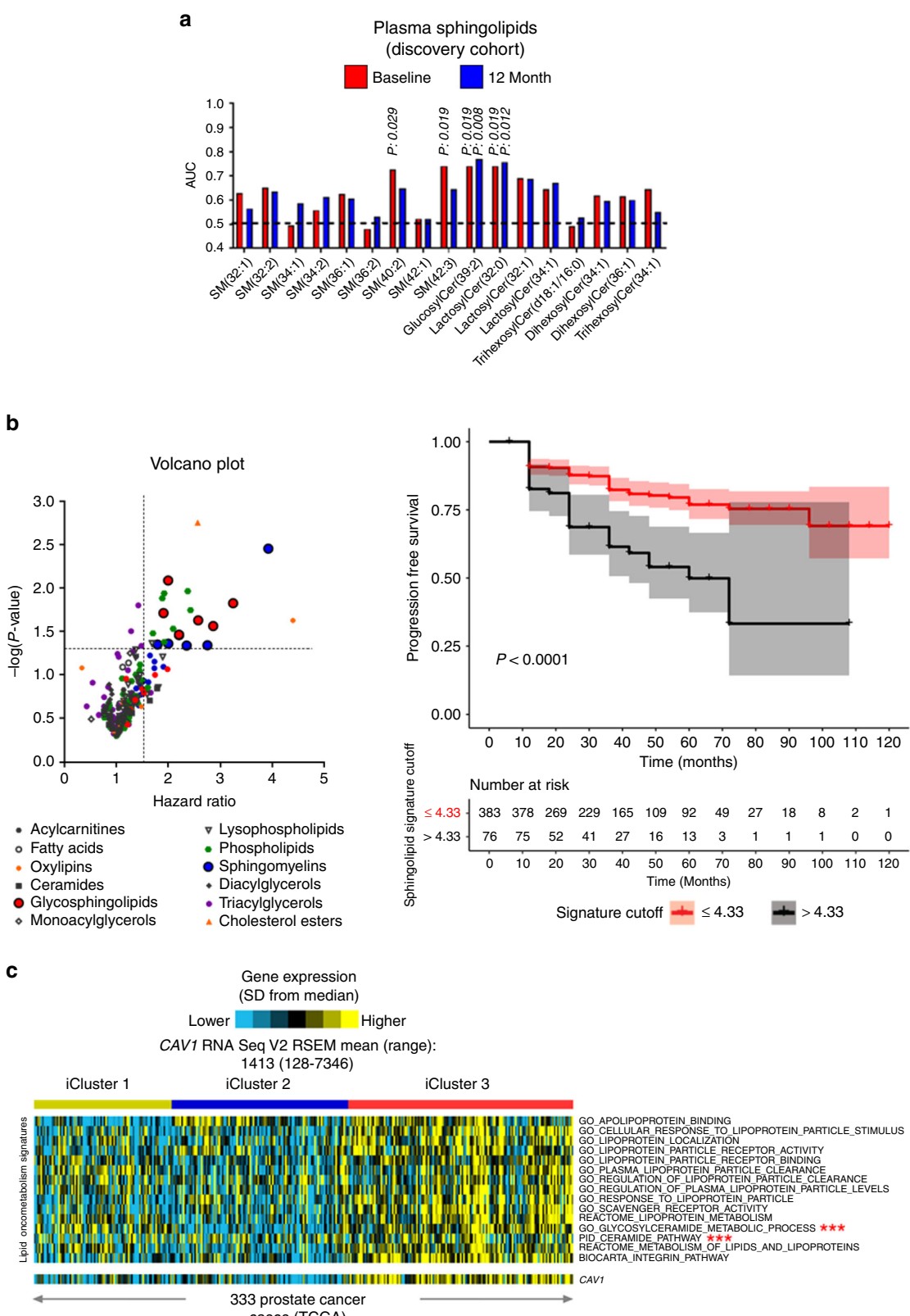

**Cav-1 is associated with a high-lipid scavenging phenotype.** To elucidate the role of Cav-1 in rewiring lipid scavenging and metabolism in prostate cancer, we next assessed whether Cav-1 was responsive to extracellular lipid availability. In comparing serum-free media with and without lipids, lipid-deprivation reduced Cav-1 protein expression in RM-9 and PC-3M prostate cancer cell lines (Fig. 2a; Supplementary Fig. 2). Notably, the presence of apolipoproteins, cholesterol, or cholesterol esters was dispensable for maintaining elevated Cav-1 protein levels, suggesting that Cav-1 is specifically responsive to the phospholipid constituents of extracellular lipid complexes (Fig. 2a; Supplementary Fig. 2). Given the above findings, we next interrogated

**Fig. 1 Plasma Cav-1 sphingolipid signatures of disease progression in patients with early-stage prostate cancer. a** Individual area under the curve (AUC) of the receiver operating characteristic (ROC) performance measurements for plasma sphingomyelins and glycosphingolipids in early stage prostate cancer patients who exhibited disease progression (defined as upgrading of Gleason score and/or increased tumor volume; $n = 16$) within 18 months after start of AS on surveillance biopsy or indolent disease (did not progress for a minimum of 5 years after start of AS; $n = 16$). Baseline, at enrollment performance is indicated by red bars; performance at 12 months is indicated by blue bars. **b** Volcano plot illustrating hazard ratios for individual plasma lipid species stratified by lipid domains in predicting disease progression using baseline plasma samples from the larger prospective cohort ($n = 459$). Hazard ratios were determined using univariate cox proportional hazard models; lipid species were treated as continuous variables. Kaplan-Meier survival curve illustrates progression free survival for participants with a plasma sphingolipid signature (plasma Cav-1 plus 6 sphingolipids) plasma sphingolipid signature levels ≤4.33 (red line) or >4.33 (black line). Optimal cut-off values for plasma sphingolipid signature were derived using two-sided log rank test statistics based methods of Contal and O'Quigley[24]. **c** Heatmap depicting gene expression profiles reflective of gene ontologies related lipid scavenging and metabolism and mRNA expression of *CAV1* in 333 prostate tumors using TCGA. Mean (range) for CAV1 mRNA levels (RNA Seq V2 RSEM) amongst the 333 prostate cancer cases was 1413 (128–7346). Ontologies marked with (***) indicate glycosylceramide metabolic process and ceramide pathway. TCGA prostate adenocarcinoma (PRAD) cases were previously classified by TCGA network using the iCluster multiplatform based method[25].

whether Cav-1 participates in lipid uptake. To this end, we assessed the ability of Cav-1 low (Supplementary Fig. 3a) LNCaP and Cav-1 positive (Fig. 2a) PC-3M and RM-9 prostate cancer cell lines to scavenge extracellular fluorescent 1,1′-dioctadecyl-3,3,3′,3′-tetramethylindocarbocyanine (DiI)-conjugated synthetic self-assembled lipid particles (SSALPs). PC-3M and RM-9 prostate cancer cells exhibited substantially higher fluorescence accumulation as compared to LNCaP (Fig. 2b).

**Cav-1 regulates glycosphingolipid biosynthesis**. We next determined whether altered Cav-1 status would manifest as a differential lipid phenotype by comparing the baseline lipid profiles of LNCaP and PC-3M cells as well as lipid profiles following respective overexpression of Cav-1, or transient knockdown of *CAV1*. Immunoblots comparing whole cell lysate Cav-1 levels in LNCaP and PC-3M cells following respective overexpression of Cav-1 or transient knockdown of *CAV1* are provided in Supplementary Fig. 3a. Relative to baseline LNCaP, baseline PC-3M cells exhibited elevated levels of triacylglycerols, cholesterol esters and lysophospholipids, whereas sphingolipids were reduced (Supplementary Fig. 3b, c). Overexpression of Cav-1 resulted in significant increases in overall levels of major lipid classes whereas Cav-1 knockdown resulted in significant reductions in phospholipids, diacylglycerols, sphingomyelins, and glycosphingolipids, particularly lactosylceramides (Fig. 2c, d). To advance these findings, we additionally evaluated the relationship between Cav-1 and enzymes central to ceramide metabolism utilizing the Broad Institute Cancer Cell Line Encyclopedia (CCLE) and TCGA gene expression datasets. For CCLE data, we stratified prostate cancer cell lines based on mean *CAV1* gene expression into either high *CAV1* expressing cell lines (log2 values >11 (range 11.01–13.61); HPrEC, DU145, PC-3) or low *CAV1* expressing cell lines (log2 values <7 (range 4.16–6.88); NCIH660, MDAPCa2B, LNCaP, VCaP, and CWR22Rv1). TCGA data on prostate adenocarcinomas was stratified into the highest or lowest *CAV1* expression quartiles to evaluate the association between *CAV1* mRNA expression and mRNA expression of genes involved in sphingolipid metabolism amongst the most differential populations. Spearman correlation analyses based on the entire TCGA prostate adenocarcinoma dataset using continuous values for mRNA expression of CAV1 and genes involved in sphingolipid metabolism are provided in Supplementary Table 6. As compared to those with low *CAV1* mRNA levels, prostate cancer cell lines and prostate tumors exhibiting high *CAV1* mRNA levels tended to also exhibit reduced mRNA expression levels for genes involved in the biogenesis of ceramides including dihydroceramide desaturase enzymes (DEGS), ceramide synthase enzymes (CERS), and sphingomyelinases (SPMDs) (Fig. 3a, b). In contrast, mRNA expression of enzymes involved in the biosynthesis of glycosphingolipids, including glucosylceramide

synthase (UCGC), and lactosylceramide synthases B4GALT5 and B4GALT6, were elevated in *CAV1*-high prostate cancer cell lines and prostate tumors (Fig. 3a, b).

Glycosphingolipids are derived through the glycosylation of ceramides. Ceramides are principally derived through three metabolic pathways: de novo, recycling, or salvaging (Supplementary Fig. 4). Ceramide biosynthesis through the salvaging pathway is mediated via the hydrolysis of sphingomyelins via sphingomyelinases (Supplementary Fig. 4). Given our observations that (1) Cav-1 is responsive to extracellular lipid availability (Fig. 2a), (2) Cav-1 associates with a high-lipid scavenging phenotype (Fig. 2b), and (3) manipulation of Cav-1 regulates glycosphingolipid biosynthesis (Fig. 2c, d), we interrogated whether uptake of extracellular sphingomyelins provides an appreciable source of ceramides and their glycosphingolipid derivatives. To this end, we treated PC-3M, RM-9, and LNCaP prostate cancer cells for 48 h with SSALPs containing sphingomyelin(d18:1/18:1)-deuterium(d) 9 and traced its biochemical fate using liquid chromatography mass spectrometry (Fig. 2e). We detected the ceramide(18:1/18:1)-d9 isotopologue in all three cell lines, whereas the glucosylceramide(18:1/18:1)-d9 isotopologue was only detected in PC-3M and RM-9 prostate cancer cell lines. Notably, the ratio of ceramide(18:1/18:1)-d9 to sphingomyelin(18:1/18:1)-d9, based on peak area, was appreciably higher in PC-3M (ratio: 0.14) and RM-9 (ratio: 0.23) as compared to LNCaP (ratio: 0.02), demonstrating an overall higher metabolic flux into ceramide biosynthesis via sphingomyelin salvaging (Fig. 2f). We did not observe the presence of oleate-d9 or ceramide(18:1/18:1-d9)-1-phosphate, indicating preferential shunting of sphingomyelin-derived ceramides into the glycosphingolipid pathway rather than hydrolysis by ceramidases or phosphorylation. These findings give direct biochemical evidence that sphingomyelins are indeed a source of ceramides and their glycosylated derivatives (Fig. 2d).

**Cav-1 facilitates turnover of mitochondrial components**. Our findings above suggest a Cav-1-associated mechanistic framework that directs ceramide pools into glycosphingolipids, consistent with our observation that glycosphingolipids, particularly lactosylceramides, are key features of the plasma sphingolipid signature (Fig. 1b). Ceramides are bioactive sphingolipids that are actively involved in mediating cell death, including induction of apoptosis through cytochrome c release from the mitochondria, as well as targeting of mitochondria to autophagosomes to elicit lethal mitophagy[26,27]. We therefore first assessed whether an association exists between Cav-1 and mitochondrial morphology in PC-3M (Cav-1 high) and LNCaP (Cav-1 low) cell lines. PC-3M cells exhibited a more branched, fused-like-mitochondrial architecture with diffuse lysosomal staining, whereas the morphology of the mitochondria and lysosomes were more punctate in LNCaP cells (Fig. 4a). Using fluorescent-labeled SSALPs

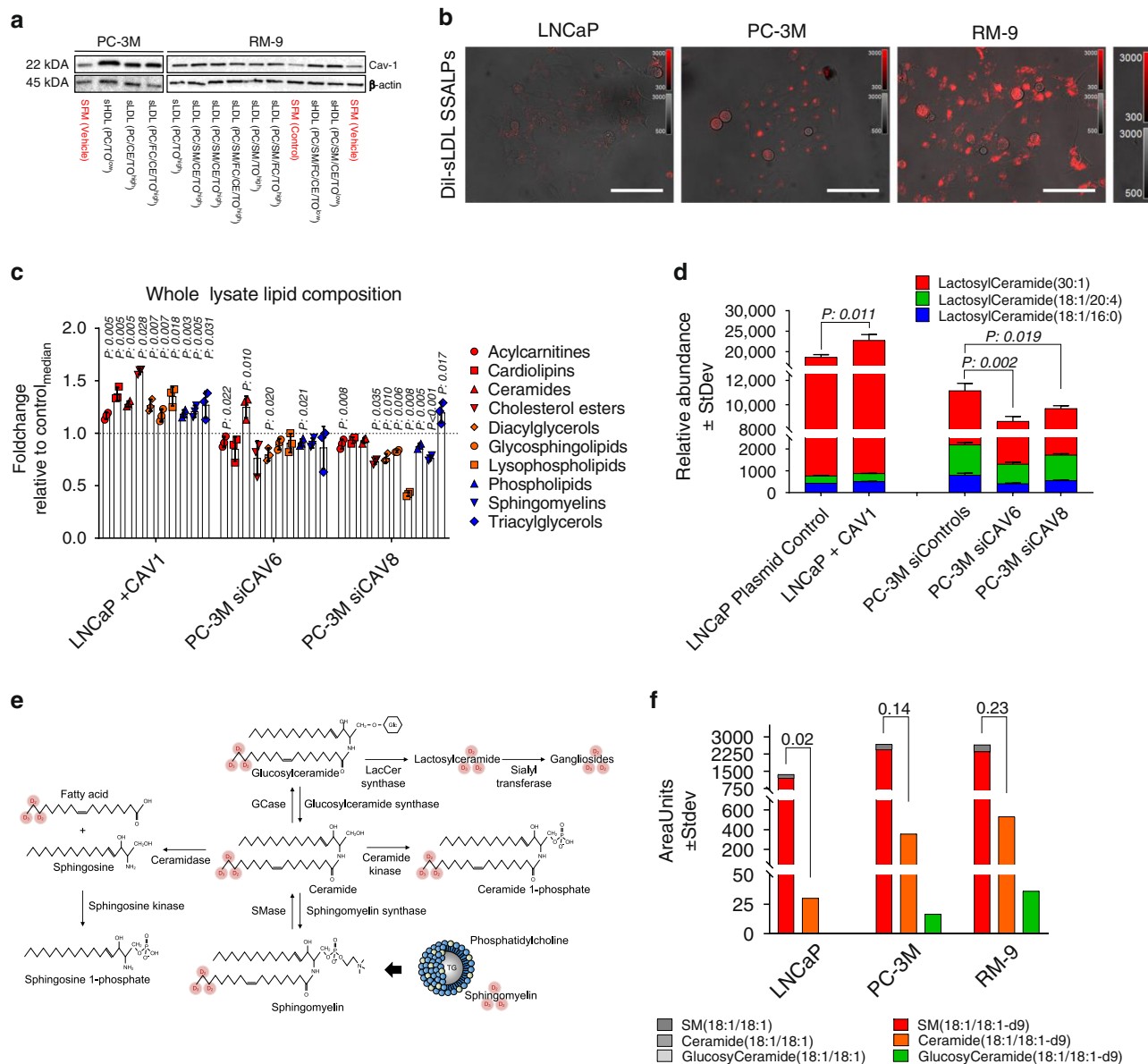

**Fig. 2 Cav-1 regulates sphingolipid metabolism. a** Immunoblot for Cav-1 in PC-3M cells following 72 h treatment with serum free media or lipid-containing serum-free media. For these experiments, synthetic self-assembled lipid particles (SSALPs) of defined lipid composition were generated and spiked into serum-free containing media. sLDL synthetic LDL-like particles, sHDL synthetic HDL-like particles, PC phosphatidylcholine, TO trioleate, CE cholesteryl oleate, FC free cholesterol. **b** Baseline assessment of Dil-SSALP uptake by LNCaP, PC-3M, and RM-9 prostate cancer cells. Cells were pretreated for 2 h with Dil-SSALPs. All images were captured via confocal and widefield microscopy using live cell imaging techniques. Scale bars represent 50 μm; LUT intensity scaling was uniform for all images (right). Digital image acquisition parameters and look up table mappings were uniformly set for all images within each respective panel. **c** Fold change (relative to median of cell line-specific control) ± StDev in lipid domains following overexpression or transient knockdown of *CAV-1* in LNCaP and PC-3M, respectively. $n = 3$ biologically independent replicates per experimental condition. For lipid domains, the aggregate intensity of individual annotated lipid species corresponding to the respective lipid domain was used. Statistical significance was determined by two-sided student *t*-test. **d** Relative abundance (area units ± StDev) of lactosylceramides following overexpression of *CAV-1* in LNCaP or transient knockdown of *CAV-1* in PC-3M. $n = 3$ biologically independent replicates per experimental condition. Statistical significance was determined by two-sided student *t*-test. Lipid abundances were normalized based on total cell number. **e** Schematic illustrating potential biochemical fates of sphingomyelin(18:1/18:1)-d9. **f** Relative abundance (area units) for sphingomyelin(18:1/18:1), ceramide(18:1/18:1) and glucosylceramide(18:1/18:1) as well as their deuterated (d9) isotopologues in LNCaP, PC-3M and RM-9 following 48 h treatment with SSALPs-enriched in sphingomyelin(18:1/18:1)-d9. $n = 2$ biologically independent replicates per experimental condition. Values presented above bar plots indicate the ratio between the ceramide(18:1/18:1)-d9 and sphingomyelin(18:1/18:1)-d9.

containing C11 TopFluor-SM, we next assessed differential trafficking of sphingomyelin in PC-3M cells following knockdown of *CAV1*. *CAV1* knockdown resulted in statistically significantly (Tukey multiple comparison test two-sided adjusted $p < 0.001$) reduced uptake of the C11 TopFluor-SM-containing SSALPs

(Fig. 4b; Supplementary Fig. 5a). Remarkably, knockdown of *CAV1* in PC-3M also resulted in the accumulation of punctate mitochondria[28–30] (Fig. 4c; Supplementary Fig. 5b, c) and a reduction in prevalence of lysosomes (Fig. 4c); these changes were accompanied by increases (Tukey multiple comparison test

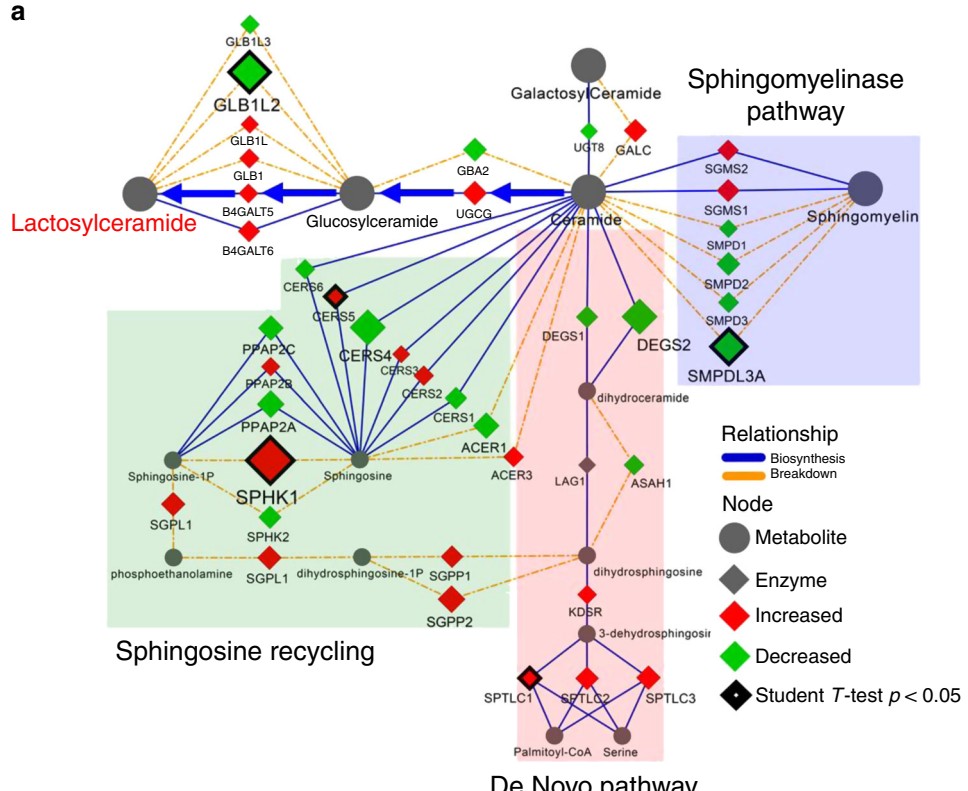

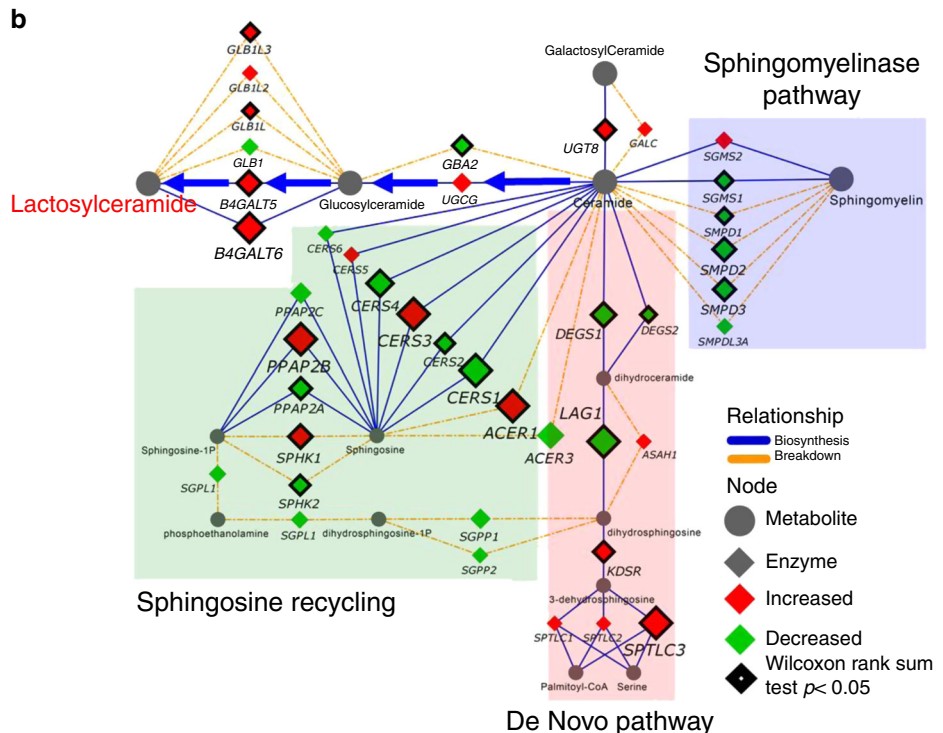

**Fig. 3 Association between gene expression of *CAV1* and enzymes of ceramide metabolism in prostate cancer.** Biochemical networks illustrating gene expression of enzymes central to ceramide metabolism in prostate cancer cell lines (**a**) and prostate adenocarcinomas (**b**) stratified by high or low *CAV-1* expression. For CCLE data (**a**), prostate cancer cell lines were stratified by mean *CAV1* mRNA expression into either high (log$_2$ mRNA range: 11.01–13.61) or low (log$_2$ mRNA range 4.16–6.88) *CAV1* expression. For TCGA data (**b**), prostate adenocarcinomas were stratified into the highest or lowest *CAV-1* expression quartiles. Spearman correlation analyses between continuous values of *CAV1* mRNA expression and gene expression of enzymes depicted in the biochemical networks for all prostate adenocarcinomas is provided in Supplementary Data 1. Node size and color reflect direction and magnitude of change (green-decreased in *CAV-1* high; red-increased in *CAV-1* high). Edges and arrows illustrate direction of biochemical reactions. Thickened black node boarders indicates statistically significant differences.

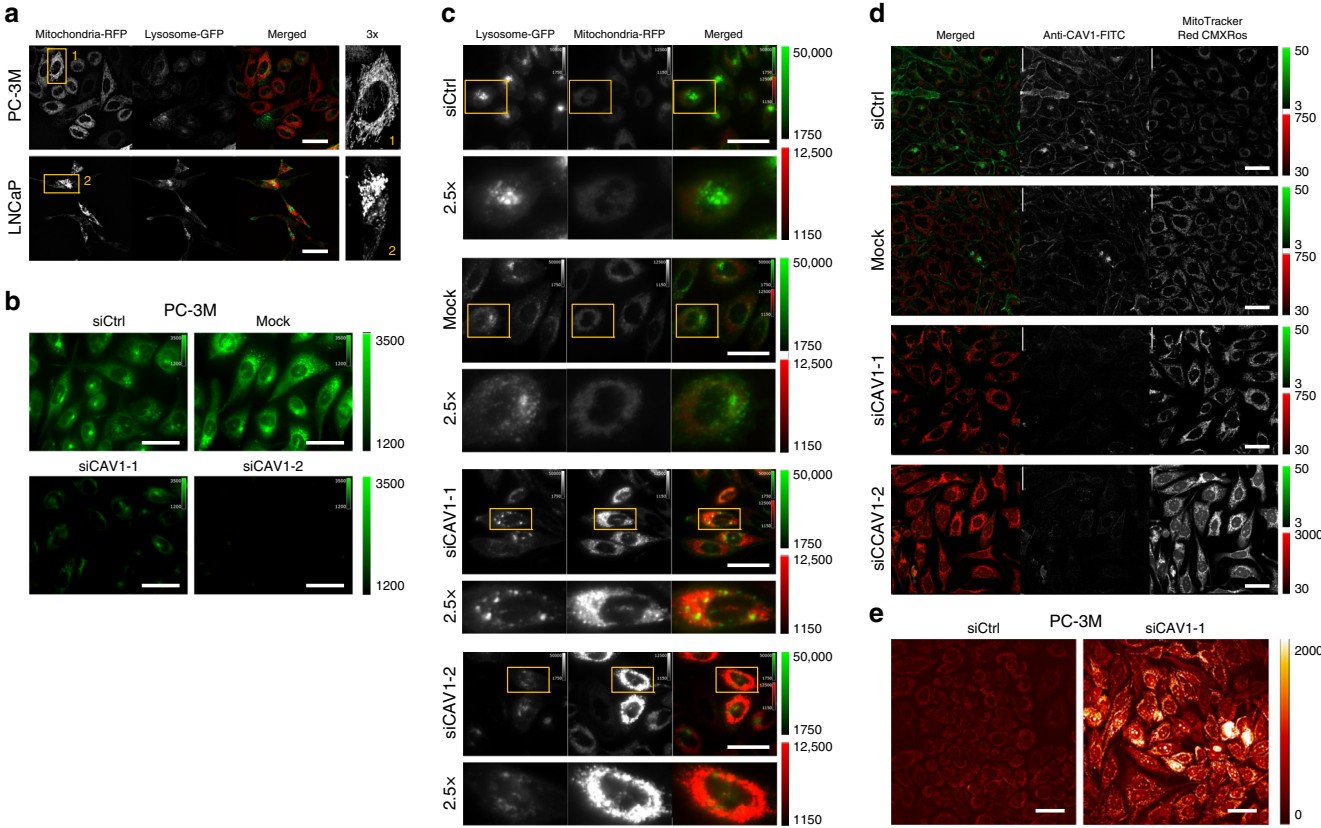

**Fig. 4 Cav-1 promotes mitochondrial turnover. a** Representative images of mitochondria and lysosomes in PC-3M and LNCaP cells transfected with CellLight Lysosome-GFP (lysosomal associated membrane protein 1) and CellLight Mitochondria-RFP (leader sequence of E1 alpha pyruvate dehydrogenase). **b** Uptake of C11 TopFluor-SM in PC-3M cells following knockdown of *CAV1*. **c** Staining for mitochondria (CellLight Mitochondria-RFP (leader sequence of E1 alpha pyruvate dehydrogenase) and lysosomes (CellLight Lysosome-GFP (lysosomal associated membrane protein 1)) in PC-3M cells following knockdown of *CAV1*. **d** Co-staining for CAV1 (FITC) and mitochondrial potential/reactive oxygen species (MitoTracker Red CMXRos[31]) in PC-3M cells following knockdown of *CAV1*. **e** Intracellular levels of reactive oxygen species assessed via CellROX Deep red in PC-3M cells following knockdown of *CAV1*. All images were captured via confocal and widefield microscopy using live cell imaging techniques. Scale bars represent 50 μm; intensity scale bars are provided next to each figure. Digital image acquisition parameters and look up table mappings were uniformly set for all images within each respective group and are indicated to the right of each image group.

two-sided adjusted $p < 0.001$) in reactive oxygen species as assessed by MitoTracker Red CMXRos[31] and CellROX Deep red (Fig. 4d, e; Supplementary Fig. 5d).

**Release of Cav-1-sphingomyelin/lactosylceramide-enriched EVs**. We have previously demonstrated that membrane associated Cav-1 is secreted by prostate cancer cells[32]. Consistent with our previous report, assessment of conditioned media (CM) confirmed detectable levels of Cav-1 in CM from PC-3M and RM-9 prostate cancer cell lines but not LNCaP (Supplementary Fig. 6a). Isolation of extracellular lipid vesicles (EVs) from CM of LNCaP and PC-3M following respective overexpression of Cav-1, or knockdown of *CAV1*, confirmed that Cav-1 is present on EVs (Fig. 5a, b). Notably, the amount of Cav-1-containing EVs was appreciably higher when culture media was supplemented with exogenous low-density lipoproteins (Fig. 5b), consistent with our earlier observations that lysate Cav-1 protein expression is responsive to extracellular lipid availability (Fig. 2a). Concordantly, in comparison to respective controls, the concentration of EV particles in conditioned media was statistically significantly higher (comparison of area under the curve two-sided student *t*-test *p*: 0.02) in LNCaP following Cav-1 overexpression whereas the number of EVs in conditioned media from PC-3M following knockdown of *CAV1* was statistically significantly reduced (comparison of area under the curve two-sided student *t*-test

$p < 0.001$) (Fig. 5c). Moreover, using density gradient fractionation, we observed that Cav-1 containing EVs were most highly enriched in a fraction within the 1.06–1.15 g mL$^{-1}$ buoyant density range of plasma high-density lipoproteins (Supplementary Fig. 6b, c), indicating that secreted Cav-1 exists in an HDL-like lipid-protein particle. Analysis of the CM lipidome of LNCaP and PC-3M following Cav-1 overexpression or *CAV1* knockdown, respectively, also indicated increased relative abundances of sphingomyelins and lactosylceramides that were dependent on Cav-1 and extracellular lipid bioavailability (Fig. 5d, e).

To determine the lipid composition and protein cargo of EVs, we performed lipidomic and proteomic analyses using mass spectrometry on EVs derived from LNCaP and PC-3M, respectively. Consistent with findings by others[33], our analyses of the EV-lipidome identified sphingolipids as well as phosphatidylcholines to be particularly enriched (Fig. 5f). Interestingly, we also identified cardiolipins, important lipid constituents of the inner mitochondrial membrane[34], to be present in prostate cancer cell line-derived EVs (Supplementary Data 3). Evaluation of the EV-proteome identified 237 and 341 (≥5 spectral abundance) high confidence proteins in LNCaP and PC-3M derived EVs, respectively. To investigate for functional aspects of protein features, we performed subcellular localization analysis based on the COMPARTMENTS localization evidence database scores [https://compartments.jensenlab.org/Search], filtering for

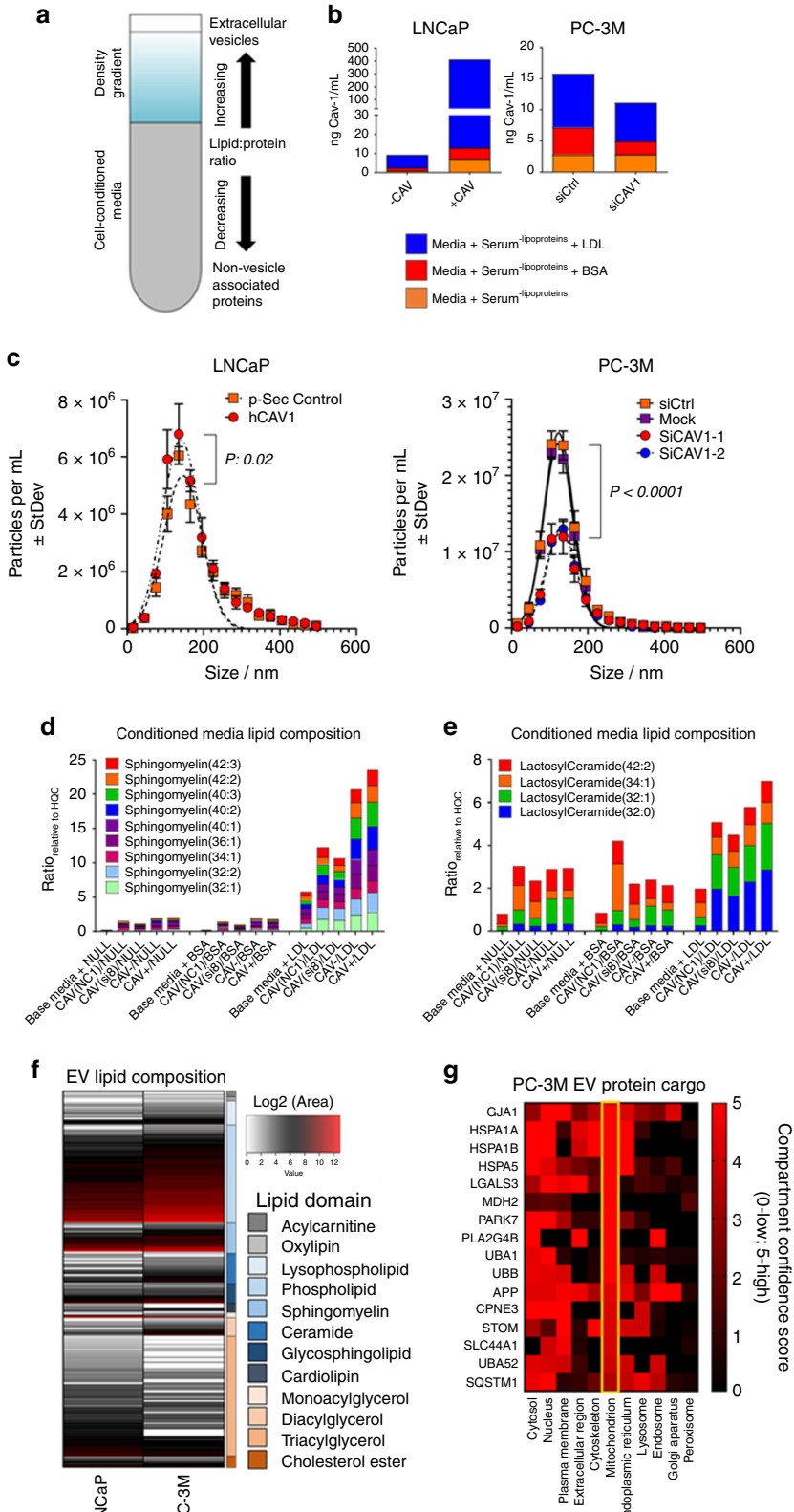

genes confidently assigned to at least one of the 11 subcellular localizations (confidence score ≥2): nucleus, cytosol, cytoskeleton, peroxisome, lysosome, endoplasmic reticulum, Golgi apparatus, plasma membrane, endosome, extracellular spaces, and mitochondrion (Supplementary Fig. 6d). Subcellular localization analyses of EV-derived protein features evidenced representation of proteins annotated as localized to the mitochondria (Fig. 5g; Supplementary Fig. 6d)[35]. IPA analysis of the 341 protein features detected in PC-3M-derived EVs revealed caveolar-mediated endocytosis and phagosome maturation as a top network, and reduced apoptosis and necrosis and increased cell movement, degranulation, and cell proliferation were predicted as top activated disease functions (Supplementary Fig. 6e). Collectively, these findings indicate that prostate cancer cells robustly secrete

**Fig. 5 Elevated *CAV1* expression facilitates secretion of Cav-1 vesicles enriched in sphingomyelins and lactosylceramides. a** Schematic of multifractionation approach. **b** Cav-1 levels (ng mL$^{-1}$) in extracellular vesicles derived from conditioned media of LNCaP and PC-3M following overexpression of Cav-1 or transient knockdown of *CAV1* in the presence or absence of BSA control or human-derived lipoproteins, respectively. **c** Particle counts (± StDev) per mL from conditioned media of LNCaP and PC-3M following overexpression of Cav-1 or knockdown of *CAV1*, respectively. $n = 1$ biological replicate and $n = 3$ technical replicates per experimental condition. Statistical significance was determined by two-sided Student *t*-test comparing the area under the curve following Cav-1 overexpression or knockdown relative to the respective scramble controls. **d**, **e** Levels of sphingomyelins (**c**) and lactosylceramides (**d**) in conditioned media of LNCaP, and PC-3M following overexpression of Cav-1 or transient knockdown of *CAV-1* in the presence or absence of BSA or human-derived lipoproteins, respectively. **f** Lipid composition of EVs isolated from conditioned media of LNCaP and PC-3M prostate cancer cells. **e** Proteins in PC-3M-derived EVs that are annotated as localized to the mitochondria based on the COMPARTMENTS localization evidence database scores. Error bars given in all figures indicate statistical significance as determined by two-sided student *t*-test.

Cav-1-containing sphingolipid-enriched EVs[33] that are enriched in a diverse repertoire of proteins including mitochondrial-associated proteins and lipids. On the basis of these findings, we surmised that Cav-1 mediated uptake of sphingomyelin (Fig. 4b), conversion of sphingomyelin to ceramide and its subsequent glycosphingolipid derivatives (Fig. 2f), and inclusion of mitochondrial proteins in the EV cargo (Fig. 5c, and Supplementary Fig. 6d) is associated with clearance of mitochondria components[36].

We speculate that the Cav-1-mediated sphingolipid trafficking apparatus reflects an adaptation of prostate cancer cells to, in part, mitigate mitochondrial toxicity.

**Shunting of ceramides into glycosphingolipids is targetable.** We considered the implications of our findings of this adaptive Cav-1-mediated glycosphingolipid mechanism and developed the hypothesis that targeting of the ceramide to glycosphingolipid conversion may represent an actionable metabolic vulnerability in prostate cancer. To test this hypothesis, we evaluated the efficacy of PDMP, PPMP, and eliglustat[37], three different inhibitors of glucosylceramide synthase (also known as UGCG; UDP-glucose: ceramide glucosyltransferase), a rate-limiting enzyme in glyco-sphingolipid metabolism[38], for reducing viability of RM-9 and PC-3M prostate cancer cells in vitro. Treatment of RM-9 and PC-3M prostate cancer cells with PDMP, PPMP, and eliglustat resulted in dose-dependent cytotoxicity (Fig. 6a). Next, we evaluated alterations in the lipidome following pharmacological inhibition of UGCG in RM-9 and PC-3M prostate cancer cells. RM-9 and PC-3M cells were challenged with PDMP, PPMP, eliglustat, or vehicle and evaluated after 6 h of treatment to capture early metabolic changes, particularly in sphingolipid metabolism, and to mitigate influence of secondary events that may cause elevations in ceramide pools[27,39,40], and reductions in GLS expression that can occur as a result of reduced cell survival[41]. Short-term (6 h) challenge of RM-9 and PC-3M prostate cells with PDMP, PPMP, or eliglustat resulted in accumulation of intracellular ceramides, acylcarnitines, lysophospholipids, and diacylglycerols and reductions in glycosphingolipids, phospholipids, and triacylglycerols (Fig. 6b, c; Supplementary Fig. 7a–c; Supplementary Data 4). Notably, the acute cytotoxic effects of eliglustat were mediated through a nonapoptotic mechanism (Supplementary Fig. 7a). Reductions in phospholipids and tria-cylglycerols coupled with elevations in their downstream cata-bolites suggested mitophagy[42,43]. Consistent with this, treatment of PC-3M cells with eliglustat increased protein expression of the mitophagy-associated markers LC3B-II (Fig. 6d). Assessment of mitochondrial morphology in PC-3M cells following acute (6 h) treatment with eliglustat indicated loss of the branched, fused-like-mitochondrial architecture, and accumulation of punctate mitochondria co-localized with the lysosome (Fig. 6e); these changes were met with increased protein expression of Parkin and PINK1, further suggesting increased mitophagy[44,45]. Previous reports have demonstrated that ceramides target autophagosomes to mitochondria to induce lethal mitophagy[26].

Notably, knockdown of *CAV1* or pretreatment with a Cav-1 specific monoclonal antibody (Cav-1mAb) further sensitized PC-3M cells to eliglustat whereas overexpression of Cav-1 in LNCaP reduced the anticancer effects of eliglustat (Fig. 6f, g).

**RM-9 tumor growth is inhibited by eliglustat in vivo.** To evaluate the efficacy of eliglustat in vivo, we subcutaneously implanted RM-9-luciferase cells into C57BL/6N mice (Fig. 7a). RM-9 cells harbor driver oncogenic *RAS* and *MYC* genes which model RAS-MAPK pathway activation and MYC-driven transcriptional activities which are associated with aggressive primary prostate cancer[46,47]. RM-9 tumor growth was suppressed by eliglustat (Fig. 7b–d). Metabo-lomics analysis of tumor tissues from all treatment groups showed that eliglustat was linked to reductions in glycosphingolipids in RM-9 tumor bearing mice (Fig. 7e). Immunohistochemical analyses of tumor tissue indicated that treatment with eliglustat statistically significantly reduced Cav-1 and PCNA staining (two-sided Wil-coxon rank sum test *p*: 0.008 and 0.001, respectively), whereas BrdU-TUNEL and mitophagy-associated marker (LC3B and HMGB1)[48,49] staining was statistically significantly increased (two-sided Wilcoxon rank sum test *p*: 0.008 for all three markers) (Fig. 7f). Notably, our results indicated that RM-9 tumors were associated with a plasma lipid signature, similar to that observed in prostate cancer patients, that included elevations in several sphin-gomyelins and glycosphingolipids (Fig. 7g). In addition, plasma Cav-1 levels tended to be elevated in RM-9 tumor-bearing mice compared to control mice (Supplementary Fig. 8a). Interestingly, plasma levels of Cav-1 and identified lipid species that are part of our Cav-1 sphingolipid signature increased upon treatment with eliglustat in RM-9 tumor bearing mice (Supplementary Fig. 8a, b). Treatment with eliglustat in RM-9 tumor bearing mice also resulted in statistically significant increases (two-sided Wilcoxon rank sum test *p*: 0.004) in plasma ceramides (Supplementary Fig. 8c), sug-gesting that the elevation in plasma Cav-1 and sphingolipids may be a consequence of cell death (Fig. 7b–d).

## Discussion

Cav-1 has been suggested to play a role in the perturbation of a number of metabolic pathways in prostate cancer cells[22,23]. Here, we delineate and experimentally demonstrate a multipoint mechanism through which Cav-1 enables rewiring of cancer cell lipid metabolism towards a program of exogenous lipid scaven-ging, altered sphingolipid metabolism, and vesicle biogenesis that intersects with mitochondrial dynamics in prostate tumors (Fig. 8). Within this Cav-1-mediated metabolic rewiring program we define discrete, coordinated activities including lipid uptake, increased conversion of extracellular sphingomyelin to ceramide and its downstream glycated derivatives that serve as lipid con-stituents of EVs, and subsequent efflux of Cav-1-sphingolipid EVs that include mitochondrial-associated proteins and lipids.

We note that our findings are consistent with substantive body of findings reported by others linking altered glycosphingolipid metabolism to cancer biological processes[50–54]. However, we

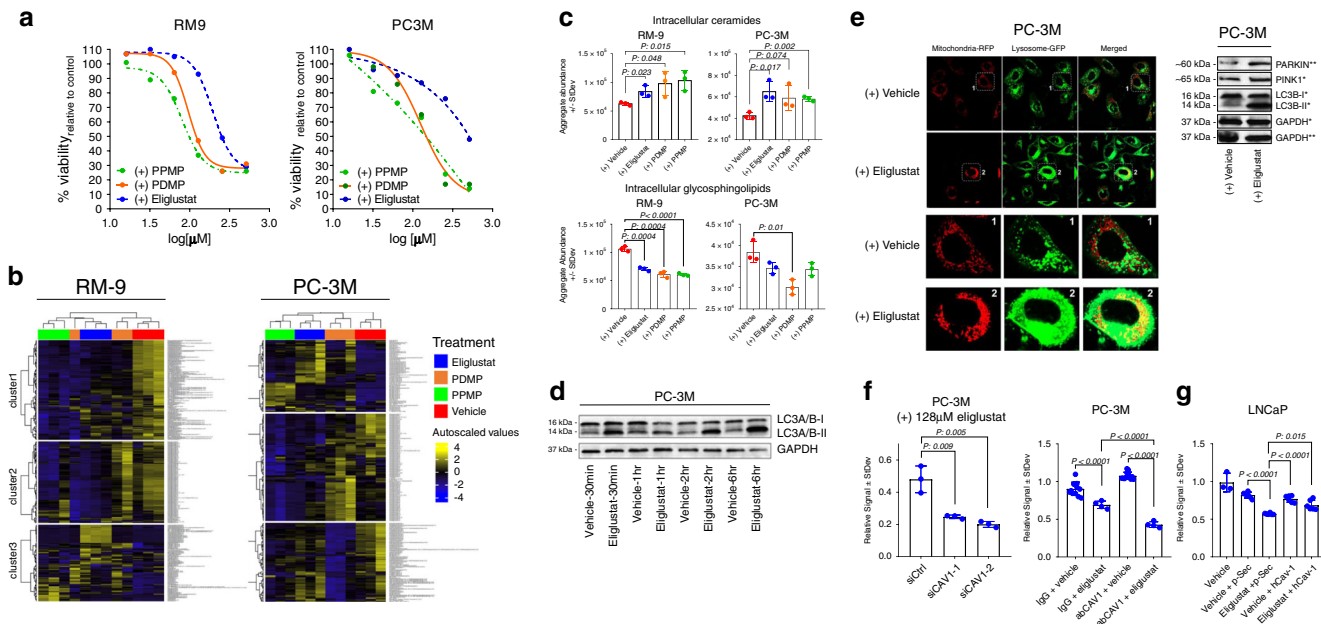

**Fig. 6 Sphingolipid metabolism is a metabolic vulnerability in prostate cancer. a** Viability curves for RM-9 and PC-3M cells following 48 h treatment with vehicle (ethanol), Eliglustat, PDMP or PPMP. **b** Unsupervised hierarchical clustering heatmaps illustrating differences in annotated lipid composition of RM-9 and PC-3M cells following 6 h challenge with either vehicle (ethanol), 128 μM Eliglustat, 64 μM PDMP, or 64 μM PPMP. Clustering was performed using Euclidean distance and Ward's method. **c** Relative abundances ± StDev of ceramides and glycosphingolipids following 6 h treatment of RM-9 and PC-3M with either vehicle (ethanol) or 128 μM Eliglustat. $n = 3$ biologically independent replicates per experimental condition. Statistical significance was determined by two-sided student t-test comparing the aggregate intensities of individual lipid species corresponding to the respective lipid domain. **d** Immunoblots for LC3B-II following treatment of PC-3M cells with either vehicle or inhibitor as indicated in b. **e** Representative confocal microscopy images for PC-3M cells following 24 h pre-transfection with CellLight Lysosome-GFP (lysosomal associated membrane protein 1) and CellLight Mitochondria-RFP (leader sequence of E1 alpha pyruvate dehydrogenase) followed by acute (6 h) treatment with either vehicle or Eliglustat (128 μM). Scale bars indicate 50 μm. LUT intensity mapping was uniform for images of all treatment conditions. Immunoblots for LC3B and mitophagy markers Parkin and PINK1 following 6 h treatment with either vehicle (left) or eliglustat (128 μM, right). Targets were probed using two different membranes; asterisks (*/**) indicate corresponding control for each target. **f** Viability (MTS assay) ± StDev of PC-3M cells treated with 128 μM eliglustat after knockdown of *CAV1* ($n = 3$ biologically independent replicates per experimental condition) or following pre-treatment with Cav-1 monoclonal blocking antibody ($n = 12$ and 4 biologically independent replicates per experimental condition for IgG and eliglustat treatment, respectively). Statistical significance was determined by two-sided student t-test. **g** Viability (MTS assay) ± StDev of LNCaP cells treated with 64 μM eliglustat following Cav-1 overexpression. $n = 6$ biologically independent replicates per experimental condition except for vehicle only treatment that consisted of $n = 3$ biologically independent replicates. Error bars given in all figures indicate statistical significance as determined by two-sided student t-test.

further acknowledge there exists considerable heterogeneity and complexity and context dependent biological functions of discrete glycosphingolipid species in cancer which we did not fully explore in the current study. Future studies will be directed at more fully explicating the cancer-supportive features of the oncometabolic network we report here, including glycosphingolipid complexity and functional roles.

Our mechanistic model posits a role for Cav-1 in mitochondrial quality control that includes clearance of mitochondria components through secretion of Cav-1-sphingolipid particles. This is consistent with an emerging view of mitochondrial maintenance as comprising different scales of quality control that include bulk elimination of mitochondria as well as targeted elimination of damaged parts of the mitochondrial network or specific mitochondrial proteins[55]. Such selective editing of mitochondria would provide an efficient response to oncogenic stress and means to evade lethal mitochondrial dysfunction. PINK1, Parkin, and LC3 have variously been found to participate in formation of mitochondrial-derived vesicles that enable quality control at the sub-organelle level by allowing mitochondria to actively shed damaged fragments in addition to known roles in targeting damaged mitochondria for autophagic elimination[29,56–59].

Importantly, this metabolic phenotype provides an additional points of intervention as well as biomarkers of disease progression and therapeutic susceptibility. Based on the finding that Cav-1 is a key mediator of ceramide metabolism and shunting into glycosphingolipid synthesis, we tested the effect of eliglustat, a selective FDA-approved inhibitor of glucosylceramide synthase intended for treatment of Gaucher's disease[60]. In an syngeneic mouse model of aggressive prostate cancer, eliglustat suppressed tumor growth, providing evidence of a targetable metabolic vulnerability to attenuate prostate tumor progression. Eliglustat challenge resulted in increased tumor staining positivity for HMGB1 and LC3B, consistent with induction of lethal mitophagy; although we do not rule out this may also be reflective of macroautophagy.

Furthermore, elucidation of the mechanistic basis underlying this lipid oncometabolism yields a Cav-1 sphingolipid signature that is detectable in blood and that may be indicative of disease progression in subjects with early stage prostate cancer who are enrolled on active surveillance. This extends the previous work, which reported that baseline plasma Cav-1 can predict disease reclassification in AS subjects, indicating that plasma Cav-1 also reflects a lipid metabolic phenotype associated with aggressive prostate cancer[4]. We have additionally identified Cav-1 as an independent prognostic tissue marker for time to disease progression in clinically confined prostate cancers[61]. Serum Cav-1 has also been shown to be elevated in men with prostate cancer

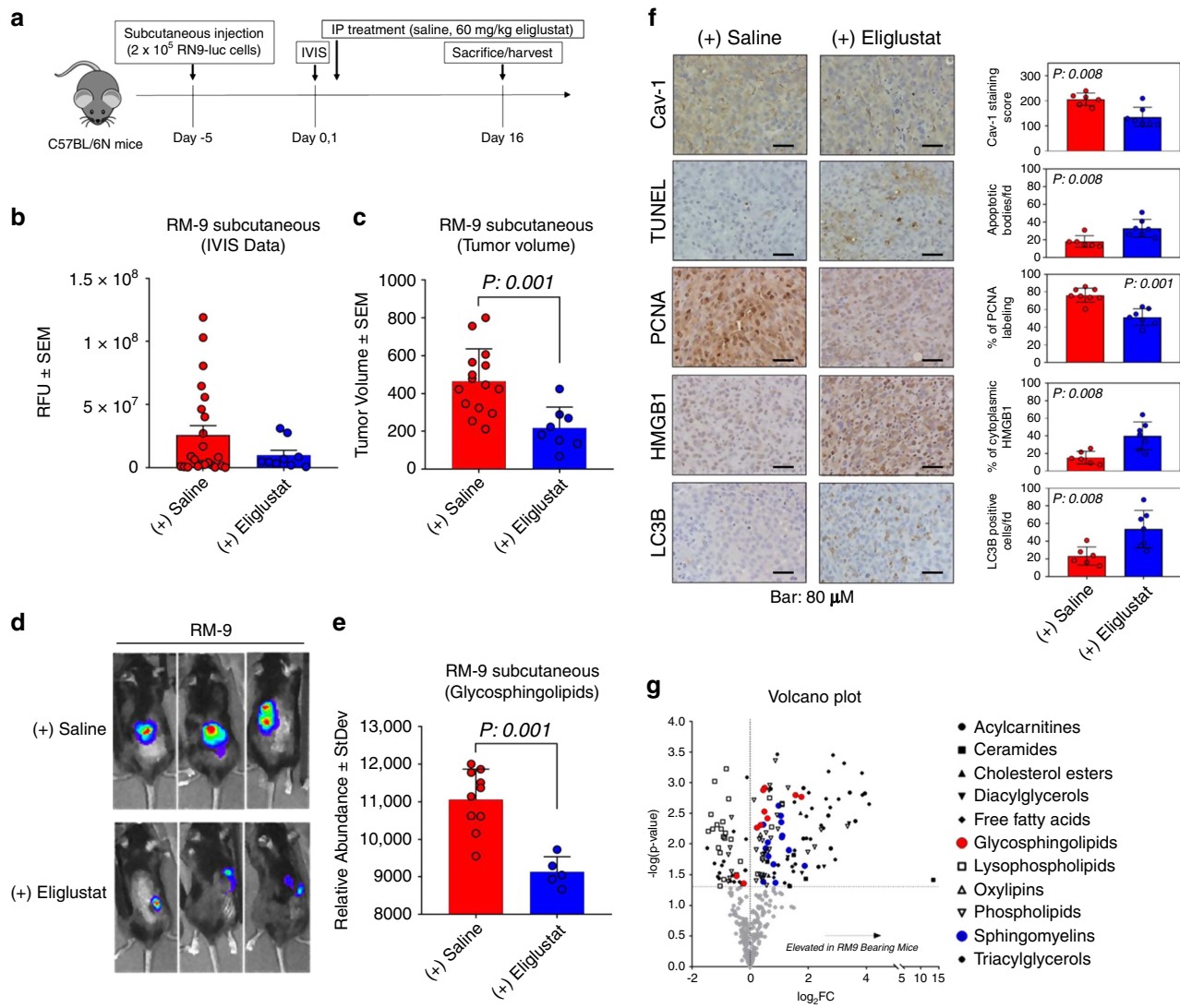

**Fig. 7 Efficacy of Eliglustat in RM-9-luciferase in vivo mouse model. a** Schematic of treatment strategy for syngeneic RM-9 C57BL/6N mouse model (Timothy C. Thompson) **b** Relative fluorescence units (RFU) ± SEM of RM-9 tumors following daily intraperitoneal injection with either saline ($n = 23$; red bars) or eliglustat (60 mg kg$^{-1}$) ($n = 9$; blue bars). **c** Tumor volume ± SEM following treatment with either saline ($n = 15$) or eliglustat (60 mg kg$^{-1}$) ($n = 8$). Statistical significance was determined by two-sided Wilcoxon-rank sum test. **d** Representative IVIS images following treatment. **e** Relative abundance (area units ± StDev) of glycosphingolipids in RM-9 tumors following treatment with saline ($n = 10$) or eliglustat (60 mg kg$^{-1}$) ($n = 5$). Statistical significance was determined by two-sided Wilcoxon-rank sum test comparing the aggregate intensities of individual lipid species corresponding to the respective lipid domain. **f** Representative immunohistochemistry staining for Cav-1, BrdU-TUNEL, PCNA, HMGB1, and LC3B in RM-9 tumors following treatment with either saline ($n = 6$ for Cav-1, BrdU-TUNEL, HMGB1, and LC3B; $n = 8$ for PCNA) or eliglustat ($n = 7$ mice). Quantitative analyses (mean ± StDev) for respective markers are provided to the right of each representative IHC section. Scale bar represents 80 µM. Statistical significance was determined by two-sided Wilcoxon rank sum test. **g** Volcano plot illustrating fold change in individual annotated lipid species stratified by lipid domain in plasma of RM-9 bearing C57BL/6N mice or control mice. Error bars in all figures indicate statistical significance as determined by Wilcoxon rank sum test.

compared to men with benign prostatic hyperplasia[62], to be associated with localized tumor burden[1], and as a possible marker to predict prostate cancer recurrence following radical prostatectomy[3]. Consistent with the lipid component of our Cav-1-sphingolipid signature, investigations of advanced cases of castration-resistant prostate cancer identified a distinct plasma lipid signature that includes sphingolipids and glycosphingolipids and that associates with poor prognosis[63], further supporting the findings we report here.

In summary, we have identified a Cav-1-sphingolipid signature as a significant independent prognostic indicator of disease progression in prostate cancer. Using multiple experimental approaches, we define a mechanism through which Cav-1 alters mitochondrial dynamics in addition to promoting cancer cell

lipid uptake and altering ceramide metabolism that results in increased glycosphingolipid synthesis and efflux of sphingolipid-enriched EVs that contain mitochondrial proteins. We further demonstrate that this activity presents a metabolic vulnerability that can be therapeutically targeted by eliglustat, a selective inhibitor of glucosylceramide synthase. Additional studies are warranted to further develop the clinical utility of these findings.

## Methods

**Active surveillance cohort study design and population.** Patients in this prospective clinical cohort included men diagnosed with localized prostate cancer and enrolled on an AS trial protocol between February 2006 and February 2014 ($n = 825$). Of these, 616 patients had at least 1 year follow-up and 491 patients had baseline plasma samples, enabling inclusion in the study.

## Lipid-centric oncometabolism

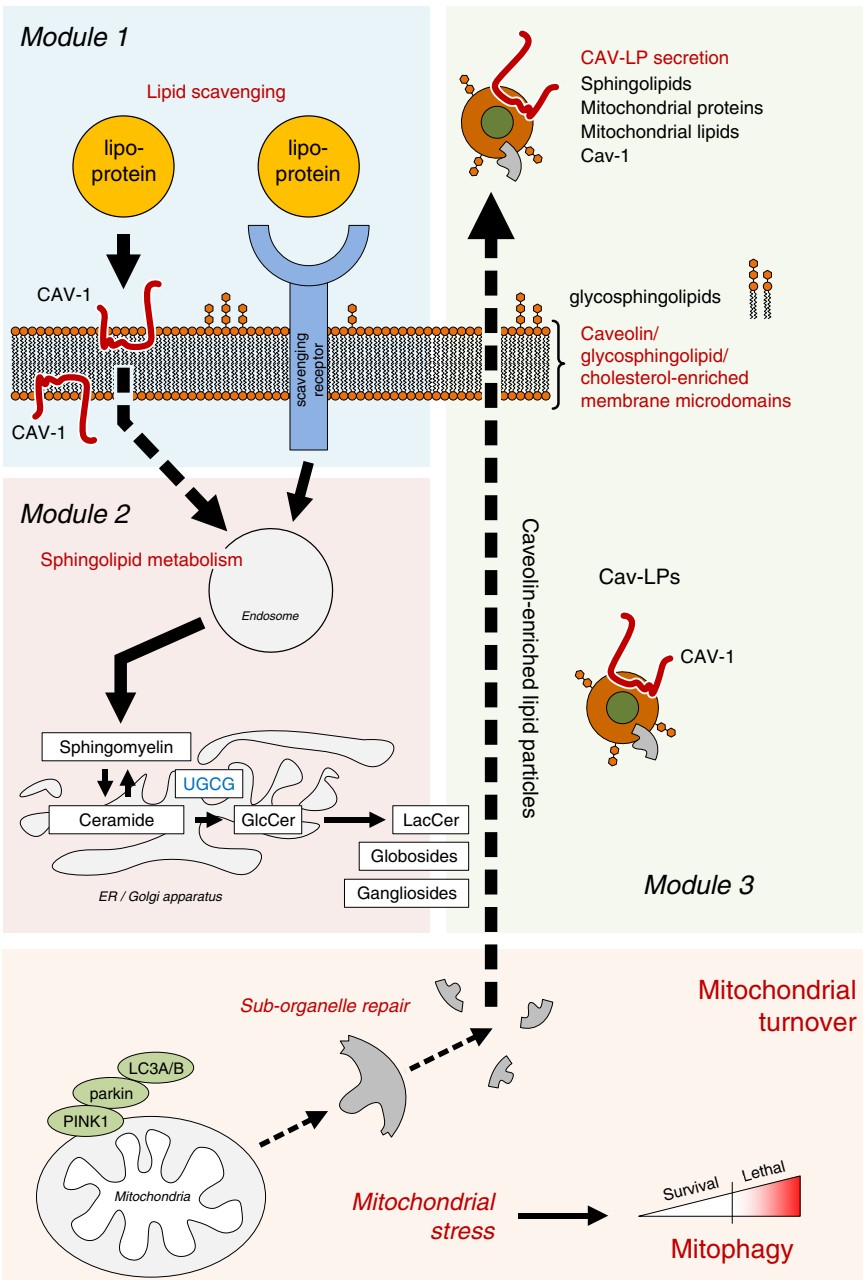

**Fig. 8 Schematic of Cav-1 mediated oncometabolism.** Essential components of this tumor supportive, lipid-centric metabolism include (Module 1) Cav-1 mediated lipid uptake, (Module 2) increased tumor catabolism of extracellular sphingomyelins, and altered ceramide metabolism that result in increased glycosphingolipid synthesis and (Module 3) efflux of circulating Cav-1-sphingolipid particles (Cav-LPs). Feed-forward system: Plasma lipid availability and lipid-enhanced signaling maintain a feed-forward circuit that reinforces upregulated Cav-1 expression and activation of metabolic programs and apparatus for lipid scavenging, glycosphingolipid synthesis and subsequent secretion of Cav-LPs. Oncometabolism: plasma-derived lipoproteins provide ready source of sphingomyelins that serve as precursors for the sphingolipid-constituents of Cav-LPs. Cav-1 supports programs involved in clearance of mitochondria components via secretion of Cav-LPs enriched with a diverse array of protein-cargo that includes mitochondria-associated proteins and lipid metabolites.

**Active surveillance protocol and outcomes assessment**. The surveillance protocol was conducted by a multidisciplinary team of urologists, radiation oncologists and medical oncologists, was approved by the Institutional Review Board, and is registered on clinicaltrials.gov (NCT00490763). Criteria including surveillance frequency and details regarding upstaging were adhered to as specified in the surveillance protocol[64]. In summary, with rare exception, patients underwent confirmatory biopsy at study entry. They were then evaluated biannually with digital rectal exam and laboratory studies (serum PSA, testosterone). All biopsies were performed using an 11-core transrectal ultrasound-guided scheme. Biopsies were repeated every 1–2 years; if one was negative, then the following year's was omitted. Treatment was discussed with patients who had an increase in tumor volume or Gleason increase, though patients who wished to remain on AS were allowed to do so if approved by their treating physician. Patients were followed until disease progression, treatment, loss to follow-up, elective removal, death, or 12/31/2016 (study censor date), whichever came first. Our outcome of interest was time to grade progression, defined as any increase in Gleason score following confirmatory biopsy. Patient characteristics are provided in Supplementary Tables 4 and 5. EDTA was used in all plasma collections and all specimens underwent a similar number of freeze/thaw cycles prior to obtaining metabolomics data. Sample ages varied, as the study began accrual in 2006 and continued over a ten year time period. Plasma was not obtained from fasted individuals.

**Chemicals**. PDMP (D-threo-PDMP) and PPMP (D-threo-PPMP) were purchased from Caymen Chemical (Cat# 10178 and Cat# 22677, respectively). Eliglustat was purchased from Abmole Bioscience Inc. (Cat# M5610). Sphingomyelin(18:1/18:1-d9) and C11 TopFluor Sphingomyelin (N-[11-(dipyrrometheneboron difluoride)undecanoyl]-D-erythro-sphingosylphosphorylcholine) were purchased from Avanti Polar Lipids (Cat# 860587 and Cat# 810250P, respectively). Triolein and cholesteryl oleate were purchased from Sigma Aldrich (Cat# T7140 and C9253, respectively); phosphatidylcholines, sphingomyelin(d18:1/16:0), and free cholesterol were purchased from Avanti Polar Lipids (Cat# 441601, 860584 and 700000, respectively). Fluorescent 3,3′-dioctadecylindocarbocyanine iodide was purchased from Life Technologies (Cat# D282). Human recombinant CAV1-6*His fusion protein was purchased from Proteintech (Cat# Ag8049).

**Preparation of synthetic self-assembled lipid particles**. Lipid stocks were prepared at $1–100$ mg mL$^{-1}$ from purified lipids dissolved in chloroform (Avanti Polar Lipids #441601, #860584, #791649 and #700000; Sigma C9253 and T7140). Lipid mixtures were standardized to 6.5 mg mL$^{-1}$ total lipids (at final concentration) and combined in ratios consistent with composition of VLDL, LDL or HDL. The solvent was removed under vacuum for 18 h (Thermo Savant SpeedVac SC250EXP). The lipid film was hydrated for 30 min at 40 °C with intermittent sonication in 0.01 M HEPES/0.15 M NaCl buffer, 1 mL per 65 mg total lipids transferred to polycarbonate tubes and subjected to five cycles of sonication, freeze (isopropanol/dry ice) and thaw (water bath, 60 °C). The resulting multilamellar vesicle suspension was sonicated (Branson Sonifier 250, 90% duty cycle, 100% power) for 12 min in a cup horn maintained at 40 °C. The lipid particle suspensions were then sterile filtered through 0.22 μm regenerated cellulose membranes and stored at $−4$ °C; for in vitro experiments, particles were used at 1:10–1:100 dilution. Particle size and yield was confirmed using nanoparticle tracking (ZetaView, Particle Metrix GmbH).

**Cell lines**. PC-3M cell line was a gift from Dr. Isaiah Fidler. PC-3M cells were seeded with 50–60% confluency at 10 cm dishes in RPMI medium containing 10% FBS. We studied three different conditions; PC-3M only; PC-3M cells transfected with Cav-1 siRNA #8 (si8); and, PC-3M cells transfected with negative control siRNA (NC). Cells were transfected with the use of Lipofectamine RNAiMAC Reagent (Invitrogen). After 24 h, medium was changed to RPMI 1640 serum free. At that point, 15 mL of medium were added to each plate instead of the usual 10 mL. After 1–2 min the extra 5 mL were taken out and stored at $−20$ °C as time zero culture medium. After 24 and 48 h culture medium and cell lysates were collected per protocol and stored at $−20$ °C.

For each condition we had triplicate dishes; one empty dish containing only medium as control; and, one dish for cell size and number counting per condition. Counting of cell size and number was performed at 24 and 48 h.

LNCaP cell line was obtained from ATCC. LNCaP cells were seeded with 50–60% confluency at 10 cm dishes in RPMI medium containing 10% FBS. We studied three different conditions; LNCaP only; LNCaP cells transfected with Cav-1 plasmid (pSECB-hCav-1); and, LNCaP cells transfected with negative control plasmid (pSECB). The day after seeding, cells were transfected with the use of X-tremeGENE HP DNA Transfection Reagent (Roche). After 24 h medium was changed to RPMI 1640 serum-free. At that point, 15 mL of medium were added to each plate instead of the usual 10 mL. After 1–2 min the extra 5 mL were taken out and stored at $−20$ °C as time zero culture medium. After 24 and 48 h culture medium and cell lysates were collected per protocol and stored at $−20$ °C. For each condition we had triplicate dishes; one empty dish containing only medium as control; and, one dish for cell size and number counting per condition. Counting of cell size and number was performed at 24 and 48 h.

Both cell lines were validated by short tandem repeat DNA fingerprinting with the AmpFℓSTR Identifier kit (Applied Biosystems) in MD Anderson's Cell Line Core Facility.

RM-9 cells were derived from Zipras/myc9-induced mouse prostate cancer and were were grown in DMEM medium containing 10% FBS[65].

**Confocal microscopy**. Mitochondria and lysosomes were imaged using BacMam CellLight Mitochondria-RFP (fused construct of the Leader sequence of E1 alpha pyruvate dehydrogenase and TagRFP) and Lysosome-GFP (fused construct of lysosomal associated membrane protein 1 and TagGFP) probes (ThermoFischer). Mitochondrial mass and potential were assessed using MitoTracker Green (Cell Signaling) and MitoTracker Red CMXRos (Cell Signaling), respectively, according to manufacturer's instructions. Total ROS was determined using CellROX Deep Red (ThermoFischer) according to manufacturer's instructions.

Imaging acquisiton and processing: Confocal images were acquired with a Nikon A1-Confocal using a Plan-Apo 60× 1.4 N.A. oil objective. Z-stacks were collected with a z-interval of 420 nm and pixels size of 100 nm using two-frame Kallman-averaging. DAPI was excited with a 405 nm laser line and emission was collected with a 450/50 bandpass emission filter. GPF was excited with a 488 nm laser line and emission was collected with 525/42 nm bandpass emission filter. mCherry was excited at 561 nm and emission was collected using a 595/50 nm emission filter, detector gains, amplifier offsets and laser power were adjusted to maximize the range of signal for each channel. All images were collected and processed with NIS-elements and deconvolution was performed using 20 interations.

All widefield fluorescence images were acquired with a Zeiss Observer Z1 and Slidebook 6.0 software (from 3i) using a Plan Apo 20× and a Plan Apo 40× oil 1.3 NA. For some images a z-stack acquision was perfomed and a nearest neighbor deconvolution (built in Slidebook 6.0) was applied.

**In vitro viability and cytotoxicity assays**. Cell viability for prostate cancer cell lines following drug treatment was determined using MTS colorimetric cell proliferation assay kits (BioVision Inc); cytotoxicity and apoptosis was determined using ApoTox-Glo Triplex Assay (Promega) according to manufacturer's instructions.

**Immunoblots**. Cells were washed three times with ice-cold 1× PBS (Gibco), and then lysed with RIPA Buffer including Protease and Phosphatase Inhibitors (Roche Lifesciences). Protein concentration was measured using the Bradford reagent (BioRad) and equal amounts of proteins (20 μg per lane) were resolved on SDS-PAGE (BioRad). Proteins were electro-transferred onto polyvinylidene-difluoride (PVDF) membrane. After blocking with 5% nonfat milk, blots were incubated with α-Cav1 (Cell Signaling, Cat# 3267S, 1:1000 dilution), GAPDH (Cell Signaling, Cat# 2118, 1:1000 dilution); autophagy marker LC3A/B-I/II (Cell Signaling, Cat# 12741, 1:1000 dilution) or mitophagy markers Parkin and PINK1 (Cell Signaling, Cat# 4211 and 6946, respectively, both at 1:1000 dilution). Uncropped original immunoblot scans are provided in Supplementary Fig. 9.

**Isolation of extracellular vesicles**. Extracellular vesicles were isolated by density gradient flotation[66] as detailed in the Supplementary Methods.

**Assessment of Cav-1 levels**. Cav-1 was measured by a direct sandwich enzyme-linked immunosorbent assay (ELISA)[4]. Characterization of the ELISA parameters, including the inter-assay and intra-assay coefficient of variability is provided elsewhere[3,62].

**Proteomic profiling of extracellular vesicles**. Proteomic profiling of extracellular vesicles was performed according to the method[66,67] detailed in the Supplementary Methods.

**Metabolomics analyses**. Metabolomics profiling was performed according to a standardized workflow[68,69]. Described in detail in the Supplementary Methods.

**Gene expression profiles**. Gene expression data for prostate cancer cell lines were obtained from Cancer Cell Line Encyclopedia (CCLE) (www.broadinstitute.org/ccle). Gene expression data and clinical data were additionally downloaded from The Cancer Genome Atlas (TCGA) network project webpage (https://tcga-data.nci.nih.gov/tcga/). Networks were visualized using cytoscape[70].

TCGA prostate adenocarcinoma (PRAD) cases were previously classified by TCGA network using the iCluster multiplatform based method[25]; only PRAD cases previously included in the TCGA network study were included in this present study. As carried out previously[71], expression values for each gene were normalized by centering them to standard deviations from the median within the TCGA PRAD RNA-seq dataset. For selected pathway-related gene sets as defined by the Molecular Signatures Database (MSigDB)[72], the average of the normalized values was computed for all genes falling under a given MSigDB gene set of interest.

**Statistics and reproducibility**. Detailed information is provided in Supplementary Methods. Log rank statistic based methods as described by Contal and O'Quigley[24] were used to determine optimal Cut-off point for plasma sphingolipid signature to distinguish subjects on AS that exhibited disease progression (defined as upgrading of Gleason score and/or increased tumor volume) from those that did not. For two class comparisons, statistical significance was determined using student t-test or Wilcoxon-rank sum test; for more than two class comparisons statistical significance was determined using one-way analysis of variance (ANOVA) multiple comparison test with Tukey HSD post hoc test to determine specific group differences unless otherwise specified. Univariate and multivariate cox proportional hazard models and construction of Kaplan–Meier survival curves were carried out using R statistical software. Using disease progression status as our binary response variables, logistic regression models with the logit link function were utilized to build a consensus signature panel.

Features that were included in this consensus signature were based on those sphingolipids that, individually, exhibited a statistically significant hazard ratio (>1.5) based off univariate Cox proportional hazard models. For multivariate Cox proportional hazard models, co-variates were included on the basis of whether they were significant during univariate analyses. Figures were generated in GraphPad Prism v7 or R statistical software.

Western blot images shown in Figs. 2a, 6d, e and Supplementary Fig. 3a report single experiments that include parallel experimental variables, multiple siRNA controls and other relevant experimental and loading controls. Micrograph images in Figs. 2b, 4a–e, 6e and Supplementary Fig. 5b show fields chosen as representative for given experimental conditions based on subjective evaluation of multiple fields

comprising ~2 cm$^2$ of cultured cell area. Comparisons are reported for images captured during a single imaging session. Image acquisition and experimental conditions were iteratively optimized with respective final experiments performed and imaged in parallel using adjacent wells of glass bottom multiwell cell imaging culture plates. Cell images reported as representative present cells with characteristics that typify those observed across the cell culture area and that are consistent with observations made during optimization experiments.

**RM-9 syngeneic mouse prostate cancer model.** Subconfluent mouse RM-9 prostate cancer cells[65] (Dr. Timothy C. Thompson, also ATCC CRL-3312), previously transduced with luciferase lentivirus, were injected into 8 week-old C57BL/6N mice (Jackson Labs, Main). Cells ($5 \times 10^4$ in 100 μL (1:1, PBS: Matrigel)) were injected subcutaneously, and 3 days after the cell injection, tumor growths were confirmed by bioluminescence and the mice were distributed into five treatment groups. On the following day, the mice received daily intraperitoneal injection of 60 mg kg$^{-1}$ of eliglustat ($n = 9$) (MedChem Express, NJ) or saline control ($n = 23$). A sample size of 9 eliglustat (case) and 23 saline control yields >99% power at a significance level ($\alpha$) of 0.05 to detect a difference in tumor volume of $\mu_1 - \mu_2 = 300$ using a one-sided two-sample $t$-test assuming that tumor volumes are normally distributed with standard deviation of 100 and 150 in eliglustat and saline groups, respectively. In addition, a separate cohort of nontumor bearing C57Bl/6N mice were treated with the same drugs regimen as the tumor bearing mice. Both tumor and non-tumor bearing groups received the treatment for 15 days, after which the tumors and the blood samples were collected. All animal experiments were conducted in accordance with accepted standards of humane animal care approved by MDACC IACUC.

**Immunohistochemistry.** Primary antibodies to Cav-1 (Santa Cruz sc-894, 1:200 dilution), PCNA (Dako M0879, 1:400 dilution), HMGB1 (Cell Signaling #6893S, 1:200 dilution) and LC3B (Sata Cruz sc-376604, 1:100 dilution) were used for immunostaining on formalin-fixed paraffin-embedded tissue slides from the RM-9 tumors. Briefly, after tissue sections were deparaffinized and rehydrated through graded alcohol, they were heated by microwave in 0.01 M citrate buffer at pH 6.0 for 20 min to retrieve antigens. After a 30-min incubation in Dako protein block, tissue sections were incubated in primary antibodies, followed by incubation in HRP polymer conjugated secondary antibody (Dako cat#K4061). The immunoreaction was visualized in DAB/H$_2$O$_2$ (Dako K3468). The specificity of immunoreactions was verified by replacing the primary antibodies with PBS. The terminal deoxynucleotidyl transferase–mediated dUTP-biotin nick-end labeling (TUNEL) technique was used to label apoptotic bodies in the RM-9 tumors with a cell death-detection kit (Millipore cat#S7100), following the manufacturer's procedure.

For immunofluorescence, PC-3M cells were fixed with ice-cold methanol followed by blocking for 1 h with KPL 10% goat serum (SeraCare). Cells were subsequently incubated overnight at 4 °C with α-Cav-1 primary antibody (Cell Signaling #3238s) at 1:400 dilution in blocking buffer (1× PBS + 1% BSA + 0.3% Triton X-100), followed by incubation in FITC-conjugated secondary antibody (BD Pharmingen # 554020) at a dilution of 1:5000.

**Generation of Cav-1mAb.** Cav-1mAb was commercially produced by conjugating peptide (LVNRDPKHLNDDVVC) with KLH as immunogen and by immunizing BALB/c mice with the conjugated peptide and then subsequently purified using Protein G affinity column.

**Reporting summary.** Further information on research design is available in the Nature Research Reporting Summary linked to this article.

## Data availability
The MS metabolomics data generated and analyzed during this study have been deposited to the NIH Common Fund's National Metabolomics Data Repository (NMDR), Metabolomics Workbench, with Project ID PR000964 (Study ID ST0014080) accessible directly via Project https://doi.org/10.21228/M83406. The MS proteomics data generated and analyzed during this study have been deposited to the ProteomeXchange Consortium via the PRIDE partner repository with the dataset identifier PXD019441. Gene expression data for prostate cancer cell lines were obtained from Cancer Cell Line Encyclopedia (CCLE). Gene expression data and clinical data were additionally downloaded from The Cancer Genome Atlas (TCGA) network project webpage and the cBioPortal public Data Portal. Networks were visualized using cytoscape. TCGA prostate adenocarcinoma (PRAD) cases were previously classified by TCGA network using the "iCluster" multi-platform based method as previously described by the Cancer Genome Atlas Research Network. Other relevant data supporting the findings of this study are available within the Article and Supplementary Information, or are available from the authors upon reasonable request.

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

## Acknowledgements

This work was supported by the generous philanthropic contributions to The University of Texas MD Anderson Cancer Center Moon Shots Program, a faculty fellowship from the University of Texas MD Anderson Cancer Center Duncan Family Institute for Cancer Prevention and Risk Assessment (J.F.F.), an Early Investigator Award from the Prostate Cancer Research Program of the US. Department of Defense Congressionally Directed Medical Research Program (W81XWH-18-1-0173) (J.R.G.), the MD Anderson NCI Prostate Cancer SPORE Grant P50 CA140388, the NCI Cancer Center Support Grant P30 CA16672, and a NIH R21 (1 R21 CA223527) (T.C.T., J.D.). C.J.C. is supported in part by NIH grant CA125123. C.B.P. is partially supported by NIH/NCI Cancer Center Support Grant P30CA016672 (Biostatistics Resource Group) and by NIH/NCI R21CA223527. Confocal microscopy in the BSRB Microscopy Facility is supported by the NIH shared instrumentation grant (1S10OD024976-01). The NMDR is supported by NIH grant U2C-DK119886.

## Author contributions

J.V., J.F.F., J.K., S.H., and T.C.T. conceived the study. J.V. and J.F.F. developed methods. J.V., J.F.F., J.R.G., Z.T., S.B., S.P., G.Y., A.F., J.M., A.P., and E.M. performed experiments and/or data collection. J.V., J.F.F., J.R.G., E.I., C.J.C., and C.B.P. performed formal data analyses. J.B.D., J.D., J.K., S.H., and T.C.T. provided resources. J.V. and J.F.F. prepared the original manuscript draft, and all authors participated in final review and editing. J.V., J.F.F., J.B.D., J.K., S.H., and T.C.T. supervised and administered the study. J.K., S.H., and T.C.T. acquired funding to support various aspects of the project.

## Competing interests

The authors declare no competing interests.
