## [Peer Review File · Nature Communications]

Reviewers' comments:

Reviewer #1 (Remarks to the Author):

This manuscript reports the association between plasma sphingolipid profile and disease progression in low risk localized prostate cancer, and attempts to link this observation to a targetable biology in the cancer cells involving scavenging and metabolism of extracellular sphingolipids. The manuscript features some interesting lipidomic profiling and sphingolipid biochemistry, but is ultimately limited by several experimental concerns and the use of laboratory models that poorly recapitulate clinical disease, particularly for early stage localized prostate cancer which is the relevant patient group in active surveillance cohorts.

The cell line models used in this manuscript are not typical of active surveillance patients or indeed most clinical prostate tumors. The only lines that express CAV1 are PC-3, RM9 and DU145, all of which are androgen receptor negative and reflect a very small proportion of very advanced or neuroendocrine prostate tumors. Virtually all early stage and low risk primary tumors express the androgen receptor and all of the AR-positive cell lines here (and seemingly most clinical tumor epithelium on Protein atlas) appear to express little to no CAV1, which casts some doubt on the mechanistic links being drawn by the authors between the plasma and cancer cell data.

Major comments:

1. Lipidomics data is not quantitative in this manuscript but rather expressed as HRs relative to historical plasma values- this is not explained. Can the authors please provide further information about how specific metabolites/lipids were identified and which standards were used? Was any interference from the presence of EDTA and/or age of the plasma samples evident in the samples detected for metabolomics? Were the plasma samples from fasted individuals?

2. I have a concern over some of the clinical terminology used in the manuscript- when considering upgrading in an active surveillance cohort, this should not be described as aggressive versus indolent prostate cancer. There is a wealth of evidence that the vast majority of even those men experiencing upgrading or other measures of disease progression will not ultimately develop clinical metastases or die from prostate cancer. Even biochemical relapse following prostatectomy has been shown to be a poor surrogate for overall survival. This aspect of the manuscript and its title needs to be described more accurately, and the results interpreted with this in mind- the signatures found are really prognostic for disease progression at best, not aggressive disease.

3. Please justify the 25%:75% cutpoint for the Kaplan-Meier survival curve in Figure 1B. Importantly, when considering an active surveillance cohort, “Survival probability” in this context (Y-axis) is misleading and should more clearly relate to disease progression or upgrading. Figure 1C is not well explained in the text and it is not evident why pathways involved in sphingolipid metabolism were not included considering the focus of the rest of Figure 1. Can the authors please also include a figure presenting the absolute expression data for CAV1 in the TCGA samples rather than relative levels? Alternatively IHC staining data from their own cohorts. This is critical to support the link between the plasma data and the mechanistic data in the rest of the manuscript. My quick impression is that the abundance is rather low in most localized tumor samples.

4. For all of the CAV1 overexpression (LNCaP) and knockdown (PC-3) data in the manuscript, the degree of overexpression or knockdown is not shown anywhere in the manuscript, figures, legends or Supplemental data. This is unacceptable and needs to be included for all major experiments.

5. Figure 2E lacks clarity and could be annotated to emphasize the main points regarding the biochemical pathways being highlighted by the authors.

6. Figure 3: CAV1 staining should be included in all of these experiments to show the extent and efficiency of knockdown in the cell populations shown- particularly important when highlighting individual cells. This may also underpin the fact that there appear to be differences in the appearance of the mitochondria in the PC-3M cells in panel A versus Panel C.

7. Figure 4: How does the total number of EVs change in response to altering CAV1 levels?

8. Figure 6: There is no information in the manuscript or figure legends regarding the numbers of mice used in the experiments. The IHC data on single tumors is inadequate in the absence of any quantitative data on the remainder of the mice in the study. It is not evident whether the effects of eliglustat on tumor growth are due to systemic effects versus a targeted effect on the tumors- was any toxicity noted? A parallel experiment to confirm this using stable knockdown of CAV1 in the RM9 cells would help distinguish between these two mechanisms.

Minor comments:

1. Certain figures within the manuscript lack clarity or are illegible. Please consider increasing font sizes and image quality for Fig 1B, 2E, 5B at a minimum.

Reviewer #2 (Remarks to the Author):

The authors of the manuscript entitled “Cav-1-Mediated Sphingolipid Oncometabolism Underlies a Metabolic Vulnerability of Aggressive Prostate Cancer” report on caveolin-1-dependent effects on the metabolome in prostate cancer by plasma from two independent cohorts of impressive size. Moreover, CCLE and TCGA transcriptomic data are incorporated throughout the study partially supporting findings. Mechanistically, three standard prostate cancer cell lines are applied throughout the study using transient silencing and overexpression of Cav-1 as well as inhibitors/ Cav-1 specific antibody. The investigation of metabolic rewiring in prostate cancer seems to be novel. The manuscript is well written, however, there are quite a few major points (especially in regards to investigations in potential mechanisms) which require a more detailed determination for current conclusions made by the authors.

1. In regards to the volcano plot shown in Figure 1B, there seems to be a general trend towards higher HR in “early disease progression” indicating that there is no single metabolite detected suggesting a smaller risk. Is this the correct interpretation and if yes, does it mean there is a generally higher abundance of detected metabolites in the DP group?

2. The authors utilize a “sphingolipid signature” to determine the prediction power for progression-free survival separating the cohort into top 25th percentile versus bottom 75th percentile, unfortunately, without providing detailed information how this arbitrary threshold was selected. Beside further information needed and without knowing the rationale for an optimal threshold, it is suggested to split the entire cohort into quartiles which might reflect prediction power closer to what has been shown in the manuscript. How does a quartile split look like?

3. The authors have accessed the TCGA prostate cancer transcriptomic data to provide further evidence of CAV-1 in prostate cancer and to study the relationship to lipid metabolism. Here, clinicopathological data accompanied with the data sets should be thoroughly investigated in regards to CAV-1/ gene sets. Has this been done? Neither the results nor the figure legends provide details on the generation of “iCluster-1 to 3”. An appropriate dendrogram on the patients and possible prostate cancer subgrouping (if there are any) might be of benefit.

4. The Western blot Figure 2B (bottom of the images) has to be improved. Here, at least a repetitive Western blot with reference protein should be shown for all three cell lines together. In addition, an appropriate quantification should be done on the images. Stating "...exhibited substantially higher fluorescence accumulation" is insufficient. In addition, the scale bar in cell culture images is missing.

5. There is no clear evidence that neither overexpression or silencing of Cav-1 in selected cell lines is functional. The Cav-1 antibody seems to work well in Western blot (Figure 2A). Thus, the overexpression and silencing could be easily demonstrated in all cell lines used throughout this study.

6. CAV1-dependent investigation of the CCLE and TCGA in Figure 2 has been done in order to study the possible association of CAV1 and metabolite enzyme-encoding gene expression. Here, the description in the text (row 189 to 196) is not obvious in the related figures. It might be true for glycosphingolipids, but not for ceramides. A more sophisticated analysis should be done. Moreover, the stratification into high or low CAV1 is insufficiently explained. Why were only the highest and lowest quartiles chosen? What are the remaining patient numbers in both groups? Is this statistically tested?

7. The authors utilize LC-MS for tracing uptake of extracellular sphingomyelins and subsequent synthesis of ceramides and glycosphingolipids in the three prostate cancer cell lines. Drawing conclusions from the cell lines based on their endogenous Cav-1 expression doesn't seem to be appropriate. Thus, the authors should show in at least 2 cell lines that their results are Cav-1 (in-)dependent by either silencing or overexpressing Cav-1. In general, comparing cell lines based on their Cav-1 expression only, does not adequately allow to address the function of Cav-1 as differences might be unrelated to the protein. Here, additional functional tests as already performed at the beginning should be consequently applied in almost all the experiments.

8. For entire figure 3, scale bar is missing and statistical evaluation is needed. The number of experiments is neither stated in the results nor figure legend.

9. "Selective" inhibitors of glucosylceramide synthase (UGCG) were applied to test a targetable metabolic vulnerability. The literature on the specificity of UGCG inhibitors is not straightforward, especially in regards to the two inhibitors used. However, it is quite evident that inhibition of UGCG does not reduce all glucosylceramide-related glycosphingolipids. Also, studies focused on UGCG are quite controversial. The authors found cytotoxicity with accompanied alteration of metabolites in 2 cell lines. Again, other non-glycosphingolipid related drugs might cause the same effect. Have other drugs been tested with focusing on identical readout? Are PPMP and PDPM working the same way

as shown for eliglustat? Does Cav-1 overexpression (e.g. LNCaP) result in enhanced resistance to eliglustat?

10. It is not clear where "... and acute (6hr) accumulation of intracellular ceramides..." (statement row 282-285) is demonstrated. There are no data on detailed expression of metabolites as stated.

11. The colors for treatment (eligustat, PDMP, PDMP, vehicle) in figure 5B should be identical.

11. Different series of glycosphingolipids play various roles in cancer. The limited number of glycosphingolipids investigated does not allow drawing conclusions for glycosphingolipids in general taken the heterogeneity of this class of biomolecules. Thus, it would be of benefit to reflect findings reported with the literature on glycosphingolipids in the context of cancer in the discussion.

Reviewer #3 (Remarks to the Author):

In this study, Vykoukal et al identify a Cav-1 and sphingolipid/glycosphingolipids and sphingomyelin) that is prognostic of prostate cancer disease progression and present a potentially interesting mechanism whereby prostate cancer cells utilize Cav-1 and increased ceramide to glycosphingo conversion to mitigate mito-toxicity potentially via increased clearance of ceramides. In addition they replicate the plasma signature of patients in tumor bearing mice.

I found the data to be quite interesting. The paper covers a lot of ground and proposes a fairly detailed mechanism. Various technical points need to be addressed, and the mechanism could be refined a bit more. Altogether this is a good mixture of clinical and basic studies. Comments are on the heavy side, but this has as much to do with our interest as it does with science.

Major comments:

1. I am somewhat confused by their overarching mechanism: The authors demonstrate nicely that tumor cells upregulate ceramide synthesis from SM, but cells must then deal with (ceramide induced) mitochondrial toxicity by shuttling out glycosphingolipids. The mechanism (as written) doesn't make sense to me, as it is a futile cycle with large bioenergetic/biosynthetic costs. If the ceramide toxicity point is important to the authors, they should test this with rescue studies (perhaps with inhibitors?). There are various other reasons why mitochondria are damaged, so I

think they can work around it. Along these lines, are mitophagy markers generally upregulated in Cav1 KD cells?

2. Figure 2G must include the abundance of total (labeled and unlabeled) species to make any claim about fluxes. The LnCAPs could have a much larger pool size (though I think results will hold). The 10-fold abundance claim about d9-ceramide is misleading without this information.

3. This important and elegant experiment would be better tested in isogenic Cav1 +/- cells (as in Fig 2C/D). As currently shown, the Cav1 expression is correlative with their flux findings. These 3 cell lines are as different as 3 PCa patients – different genetic backgrounds, etc.

4. The impact of Cav1 modulation on cell growth should be shown. This also puts the flux and metabolomics data in better context. Is this dependent on lipids in the medium? It would provide the reader with more confidence in their mechanism and data in Figure 3. I like the mitochondria efflux concept, but this is not supported enough by the data provided.

5. Are significant amounts of mitochondrial DNA or protein present in EVs? Not +/- Cav1 (this will certainly be the case), instead the authors should compare vesicular mitochondrial pools to total to determine whether this flux is appreciable.

6. If Cav-1 protects against mito toxicity, is overall mito function altered with Cav-1 KD such as oxygen consumption etc.?

7. The clinical data is impactful, but none of the metabolites in question were statistically significant in their initial analysis (as the authors note). I am fine with the subsequent grouping/signature that focuses on ceramide metabolism, but these trends should be considered in the context of other clinical parameters. Could obesity or BMI be a driver of the dyslipidemia? Most plasma lipids are on liver-derived vLDL particles and thus reflects hepatic metabolism more than the tumor. On that front, it would be helpful to quantify the amount of sphingolipids in patient EVs vs vLDLs to know what they are observing in the clinical samples.

Minor points:

1. It's not my area (making it all the more important), but the FL image results weren't clear to me initially. I understand the point from the text and see the image, but it could be presented better. Some quantitation of FL imaging would allow the authors to show the highlight the difference to the reader more clearly.

2. How was the lipid abundance in figure s3, 2c-d, 2g normalized, was this to cell number or protein?

3. Authors note "Cav-1-mediated clearance of 'damaged' mitochondria", then would you expect less LC3BII with eliglustat?

4. Please add a scale bar to 6f images

5. Please plot individual values on 6 b-c so that the biological variation is clear.

Reviewer #1 (Remarks to the Author):

This manuscript reports the association between plasma sphingolipid profile and disease progression in low risk localized prostate cancer, and attempts to link this observation to a targetable biology in the cancer cells involving scavenging and metabolism of extracellular sphingolipids. The manuscript features some interesting lipidomic profiling and sphingolipid biochemistry, but is ultimately limited by several experimental concerns and the use of laboratory models that poorly recapitulate clinical disease, particularly for early stage localized prostate cancer which is the relevant patient group in active surveillance cohorts.

The cell line models used in this manuscript are not typical of active surveillance patients or indeed most clinical prostate tumors. The only lines that express CAV1 are PC-3, RM9 and DU145, all of which are androgen receptor negative and reflect a very small proportion of very advanced or neuroendocrine prostate tumors. Virtually all early stage and low risk primary tumors express the androgen receptor and all of the AR-positive cell lines here (and seemingly most clinical tumor epithelium on Protein atlas) appear to express little to no CAV1, which casts some doubt on the mechanistic links being drawn by the authors between the plasma and cancer cell data.

We thank the Reviewer for these comments and insightful questions. With regard to the relationship of the cell line models used for our experiments and clinical active surveillance patients, although our discovery work exploited the differences between indolent and actively progressing early prostate cancer we did not intend to specifically model this disease state. However, we do believe that exploration of the biological differences between indolent and progressing early prostate cancer allowed us to identify the metabolic transitions that are manifest in the plasma Cav-1-sphingolipid signature, and that investigation of this signature in turn, led to the mechanism-based, lipid metabolic pathway we describe. We do not propose the Cav-1-regulated lipid metabolic signaling pathway we report is exclusively related to a particular prostate cancer clinical state, but do believe our report will lead to further clinically relevant studies that will inform biomarkers and therapy approaches for progressing, clinically significant disease. We have carefully reviewed our manuscript to exclude ambiguous statements in this regard.

We would also like to clarify that RM-9 mouse prostate cancer cells are androgen receptor positive and castration responsive in vitro and in vivo ^{1, 2}. We believe that this cell line and similarly derived “RM lines” recapitulate many important aspects of clinical prostate cancer and, importantly can be easily adapted to in vivo tumor growth in syngeneic models.

With regard to the “little to no CAV1 expression” in clinical tumor epithelium—in addition to the comments above which are relevant to this concern—we would also like to relate that the immunostaining methods used for Cav-1 detection, and the associated analytical approaches are critical for analysis of Cav-1 expression in clinical samples. In our experience, Cav-1-specific antibodies must be carefully validated and optimized. Importantly, Cav-1 expression in low grade tumors is focal and Cav-1 expression scoring needs to incorporate consideration of this biological activity by using “Hot Spot” analysis or other approaches to analyze large numbers of tissue sections from a large number of cases to accurately assess Cav-1 positivity in prostate cancer epithelial cells ^{4, 5, 6}. We do not intend these comments to reflect in a negative

way on the validity of the information included in the Protein Atlas, but instead wish to relate the considerations that have informed many studies of Cav-1 expression in early prostate cancer samples. With regard to the mechanistic links between the plasma and cancer cell data that we report, we believe our experimental results are consistent with our clinical data, and overall our results are consistent with those included in numerous publications from colleagues throughout the world. We further speculate that the Cav-1 regulated lipid metabolic pathway may involve cellular recruitment of newly synthesized Cav-1 into trafficking functions that ultimately result in inclusion in EVs that are eliminated from the cell. In early, clinically progressing prostate cancer, these activities may result in relatively low levels of Cav-1 that are retained in the cell. This concept is purely speculative, and we intend to pursue additional trafficking, and tissue-based studies that could potentially inform the relationship between intrinsic vs secreted Cav-1 in early prostate cancer. We appreciate the Reviewer's thoughtful consideration of our work and hope we have satisfactorily addressed the concerns of this Reviewer.

Major comments:

1. Lipidomics data is not quantitative in this manuscript but rather expressed as HRs relative to historical plasma values- this is not explained. Can the authors please provide further information about how specific metabolites/lipids were identified and which standards were used? Was any interference from the presence of EDTA and/or age of the plasma samples evident in the samples detected for metabolomics? Were the plasma samples from fasted individuals?

EDTA was used in all plasma collections and all specimens underwent a similar number of freeze/thaw cycles prior to obtaining metabolomics data. Sample ages varied, as the study began accrual in 2006 and continued over a ten year time period. Plasma was not obtained from fasted individuals. Note that relevant covariables, including age, were included in our multivariable cox proportional hazard models; results indicated that our identified sphingolipid signature remained statistically significantly associated with increased risk of disease progression (HR: 2.70 (95% CI: 1.75-4.16), 2-sided P: <0.001) (**Supplementary Table S3**).

The Reviewer is correct that the lipidomics data is not presented in absolute quantitative terms. Rather than employing individual standards and absolute quantitation for each potential analyte, our untargeted metabolomic workflow yields relative quantitation of identified metabolites measured against a historical reference standard. We note that our efforts were part of a discovery phase to identify signatures that may identify those subjects at risk of disease progression (defined as upgrade in GS and/or tumor volume) and that the hazard ratios were computed by comparing (relative) quantified plasma levels of metabolites among the respective groups.

Specifically, in relation to data processing of metabolomic data, untargeted metabolomic analyses were conducted on 491 plasma samples. To correct for injection order drift, each feature was normalized using data from repeat injections of quality control samples collected every 10 injections throughout the run sequence. Measurement data were smoothed by Locally Weighted Scatterplot Smoothing (LOESS) signal correction (QC-RLSC) as previously described⁷. Feature values between quality control samples were interpolated by a cubic spline. Metabolite values were rescaled by using the overall median of the historical quality control peak areas across all samples. Consequently, values are reported as a ratio relative to a

historical reference sample, thereby enabling batch-to-batch alignment and mitigating potential analytical bias.

Annotations were determined by matching accurate mass and retention times using customized libraries created from authentic standards and/or by matching experimental tandem mass spectrometry data (particularly in relation to lipids) against Lipid Blast, NIST MSMS or HMDB v3 theoretical fragmentations. In the context of lipids, we additionally considered retention time patterns that are characteristic to specific lipid classes⁸.

2. I have a concern over some of the clinical terminology used in the manuscript- when considering upgrading in an active surveillance cohort, this should not be described as aggressive versus indolent prostate cancer. There is a wealth of evidence that the vast majority of even those men experiencing upgrading or other measures of disease progression will not ultimately develop clinical metastases or die from prostate cancer. Even biochemical relapse following prostatectomy has been shown to be a poor surrogate for overall survival. This aspect of the manuscript and its title needs to be described more accurately, and the results interpreted with this in mind- the signatures found are really prognostic for disease progression at best, not aggressive disease.

We appreciate the Reviewer's thoughtful feedback and suggestions with respect to the clinical context of our study. One of the greatest challenges in studying localized prostate cancer or deciding when to offer curative treatment to patients on active surveillance is determining the presence of disease progression or potentially life-threatening/clinically-meaningful disease. While we agree that measures of progression based on pathologic Gleason score are imperfect and do not necessarily represent life-threatening prostate cancer, we would put forth that it is a conservative, accepted measure of an increased risk of metastasis and disease-related mortality when compared to Gleason 6 disease. This fact additionally holds true in a surveillance population, as studies of the Sunnybrook surveillance cohort (which included a significant number of Gleason 7 cancers on enrollment) demonstrate that Gleason 7 disease is associated with an increased risk of metastasis and disease-specific survival and decreased overall survival⁹. However, we acknowledge the Reviewer's criticism of our terminology as valid and have altered the wordage in the manuscript to reflect that our signature is associated with disease progression (defined as an upgrade in Gleason Score and/or tumor volume on surveillance biopsy) and we have removed references to 'aggressive' disease.

In relation to model systems, particularly mouse model systems, that emulate localized, low risk prostate cancers as observed in men on active surveillance, do not readily exist. Despite these limitations, we would like to emphasize that there is good concordance between the model systems described in the manuscript and metabolic features observed in the plasmas of men with prostate cancer on AS. We additionally note that the RM-9 prostate cancer cell line expresses the androgen receptor and is androgen responsive in vitro and in vivo^{1,2}.

3. Please justify the 25%:75% cutpoint for the Kaplan-Meier survival curve in Figure 1B.

We acknowledge this critique as valid and have systematically revisited our statistical treatment of the data to address this concern. This has rendered our analyses more straightforward, and we appreciate this constructive criticism.

We have now included univariate and multivariate Cox proportional hazard models to assess the association between plasma Cav-1-sphingolipid signature scores with progression

free survival in AS subjects. Age, 5-alpha reductase treatment, and baseline tumor volume were included as co-variables in multivariate Cox proportional hazard models. Co-variables included in the model were selected using a backward stepwise method (likelihood ratio). Results are provided in **Supplementary Table S3**.

In order to find the cut-off point for the covariate that gives the largest difference between individuals in the two already defined groups, we used the method described by Contal and O'Quigley (1). Using log rank statistic-based consideration of the groups defined by cut-off we have:

$$S_k = \sum_{i=1}^D [d_i^+ - d_i \frac{r_i^+}{r_i}]$$

Where D is the total number of distinct events (disease progression (DP)), d_i is the total number of DP at each event time (t_i), d_i^+ is the total number of DP when the Cav-1-sphingolipid signature value is bigger than the cut-off point. r_i and r_i^+ also define as the total number at risk for all Cav-1-sphingolipid signature values and Cav-1-sphingolipid signature values larger than cut-off point respectively.

We calculated S_k for all possible cut point in Cav-1-sphingolipid signature column and the estimated cut-point is the value that yields the maximum S_k . In our analysis, the maximum value of S_k is at the top 16.4% of Cav-1-sphingolipid signature values. In another word, top 16.4% of Cav-1-sphingolipid signature values are in the high-risk group and other 83.6% are in the low-risk group.

In order to calculate the p -value of this test we used the following formula and it gives the value of 0.009. It suggests that the Cav-1-sphingolipid signature level highly relates to progression free survival.

$$p\text{-value} \approx 2\exp(-2Q^2)$$

where:

$$Q = \frac{\max |S_k|}{s\sqrt{D-1}}$$

and

$$s^2 = \frac{1}{D-1} \sum_{i=1}^D \left\{ 1 - \sum_{j=1}^i \frac{1}{D-j+1} \right\}^2$$

We have added this information to the main text as well as the supplementary material.

Importantly, when considering an active surveillance cohort, “Survival probability” in this context (Y-axis) is misleading and should more clearly relate to disease progression or upgrading.

We thank the Reviewer for catching this error, we have corrected the KM curve to display “Progression free survival”.

Figure 1C is not well explained in the text and it is not evident why pathways involved in sphingolipid metabolism were not included considering the focus of the rest of Figure 1. Can the authors please also include a figure presenting the absolute expression data for CAV1 in the TCGA samples rather than relative levels? Alternatively IHC staining data from their own cohorts. This is critical to support the link between the plasma data and the mechanistic data in the rest of the manuscript. My quick impression is that the abundance is rather low in most localized tumor samples.

We have expanded upon Figure 1C in the text of the manuscript. We note that ceramide metabolism and glycosylceramide metabolic pathways are included in these analyses. We have now highlighted the glycosylceramide metabolic pathway and ceramide metabolism in figure 1C with ***.

We have now additionally included the mean (range) of CAV1 mRNA expression values in Figure 1C (CAV1 RNA Seq V2 RSEM mean (range): 1413 (128-7346)).

4. For all of the CAV1 overexpression (LNCaP) and knockdown (PC-3) data in the manuscript, the degree of overexpression or knockdown is not shown anywhere in the manuscript, figures, legends or Supplemental data. This is unacceptable and needs to be included for all major experiments.

We have now included immunoblots for Cav-1 in LNCaP and PC-3M following overexpression or knockdown of Cav-1, respectively. These are provided in **Supplementary Figure 3A.**

5. Figure 2E lacks clarity and could be annotated to emphasize the main points regarding the biochemical pathways being highlighted by the authors.

We have modified figure 2E (now **Figure 3A-B**) to better emphasize the main findings and to improve its legibility. We have additionally included spearman correlation analyses between continuous values of CAV1 mRNA expression and gene expression of enzymes depicted in the biochemical networks for all prostate adenocarcinomas. These results are provided in **Supplementary Table S6.**

6. Figure 3: CAV1 staining should be included in all of these experiments to show the extent and efficiency of knockdown in the cell populations shown- particularly important when highlighting individual cells. This may also underpin the fact that there appear to be differences in the appearance of the mitochondria in the PC-3M cells in panel A versus Panel C.

We agree that characterization of Cav-1 expression and functional effects at the single cell level may yield additional insights, nevertheless such characterization is technically challenging. In

the present study we explored the relationship between Cav-1 expression and altered lipid metabolism along multiple lines of inquiry. Investigations of cellular lipid uptake and trafficking, mitochondrial morphology, and ROS accumulation were conducted in parental and Cav-1 modulated prostate cancer cell lines using live cell confocal imaging to facilitate optimal capture and assessment of these dynamic processes in viable, metabolically active cell models. Concurrent assessment of Cav-1 expression in these cells would necessitate permeabilization, fixation or other manipulation that is incompatible with live cell imaging techniques. Related live cell investigations in our laboratory have revealed that treatment with fluorescent anti-Cav-1 antibody alters Cav-1 membrane localization and trafficking; Cav-1-fluorescent protein fusion constructs have been reported to introduce similar confounding effects.

7. Figure 4: How does the total number of EVs change in response to altering CAV1 levels?

We have now included experimental data that overexpression of Cav-1 in LNCaP increases the total number of EV particles in conditioned media whereas knockdown of CAV1 in PC-3M cells reduces the total number of EV particles. This data is provided in Figure 5C.

8. Figure 6: There is no information in the manuscript or figure legends regarding the numbers of mice used in the experiments. The IHC data on single tumors is inadequate in the absence of any quantitative data on the remainder of the mice in the study. It is not evident whether the effects of eliglustat on tumor growth are due to systemic effects versus a targeted effect on the tumors- was any toxicity noted? A parallel experiment to confirm this using stable knockdown of CAV1 in the RM9 cells would help distinguish between these two mechanisms.

We have now included the number of animals used per experimental arm. This information can be found in the figure legend for the respective data (Figure 7B-C) as well as the methods section of the manuscript. We have now additionally provided quantitative analyses for IHC staining (provided to the right of each representative IHC section in Figure 7F. Scale bars represent 80µM. Statistical significance was determined by 2-sided Wilcoxon rank sum test.)

The Reviewer’s inquiry regarding direct- or indirect- effects of eliglustat is valid. We note that there was no statistically significant difference in body weight following eliglustat treatment in comparison to vehicle control (2-sided Student T-test p: 0.75).

Although we do not rule out non-specific effects of eliglustat, long-term adverse event profiling from four completed trials of oral eliglustat in adults with Gaucher disease type 1 demonstrated favorable safety profiles with treatment-related adverse events being mild or moderate, transient, and occurring only once per patient¹⁰.

Based on our IHC analyses, tumor expression of the proliferation marker PCNA was statistically significantly reduced upon eliglustat challenge, whereas staining for TUNEL, an apoptotic marker, and autophagy related markers LC3B and HMGB1 were statistically significantly elevated (**Figure 7F**) suggesting direct anti-tumor effects.

With regard to the use of stable knockdown of CAV1 in the RM-9 cells to distinguish between direct and indirect effects of eliglustat, we agree that this direction of investigation is likely to yield additional insights. On the basis of our experimental results, which have been reported in our previous publications and in publications from other groups, we know that in prostate cancer xenograft models Cav-1 is taken up from circulation, and/or from cells within the tumor microenvironment, and that this exogenous Cav-1 is functional^{2, 11, 12, 13}. Because of this biological complexity, in vivo experiments that test tumor cell CAV1 KD or deletion are difficult to interpret and must be controlled for exogenously active Cav-1—if the intent is to analyze the effects of intrinsic Cav-1. We are very interested in the roles of intrinsic vs extrinsic Cav-1 in prostate cancer cells within the context of the novel experimental results included in our current manuscript, and the Reviewer has raised very important questions related to this point, which we intend to pursue in future studies. However, respectfully, we believe that these experiments are beyond the scope of the current study.

Minor comments:

1. Certain figures within the manuscript lack clarity or are illegible. Please consider increasing font sizes and image quality for Fig 1B, 2E, 5B at a minimum.

We have increased the size and image quality for the respective figures.

Reviewer #2 (Remarks to the Author):

The authors of the manuscript entitled “Cav-1-Mediated Sphingolipid Oncometabolism Underlies a Metabolic Vulnerability of Aggressive Prostate Cancer” report on caveolin-1-dependent effects on the metabolome in prostate cancer by plasma from two independent cohorts of impressive size. Moreover, CCLE and TCGA transcriptomic data are incorporated throughout the study partially supporting findings. Mechanistically, three standard prostate cancer cell lines are applied throughout the study using transient silencing and overexpression of Cav-1 as well as inhibitors/ Cav-1 specific antibody. The investigation of metabolic rewiring in prostate cancer seems to be novel. The manuscript is well written, however, there are quite a few major points (especially in regards to investigations in potential mechanisms) which require a more detailed determination for current conclusions made by the authors.

1. In regards to the volcano plot shown in Figure 1B, there seems to be a general trend towards higher HR in “early disease progression” indicating that there is no single metabolite detected suggesting a smaller risk. Is this the correct interpretation and if yes,

does it mean there is a generally higher abundance of detected metabolites in the DP group?

The Reviewer is correct that, in general, plasma sphingolipids tend to be elevated in plasmas of subjects presenting with disease progression as opposed to a single metabolite. We further note that each of these metabolites are biochemically linked suggesting a coordinate signal. Expectantly, lipids belonging to the specific lipid classes tend to be highly correlated.

We note that the coordinate signal, in our opinion, is an important consideration given that the number of possible unique lipids belonging to a lipid class, e.g. sphingomyelins, can be quite large due to variations in the fatty acyl chains. Thus, general trends for a lipid class towards higher HR in early disease progression mitigates risk of false positives due to randomness.

2. The authors utilize a “sphingolipid signature” to determine the prediction power for progression-free survival separating the cohort into top 25th percentile versus bottom 75th percentile, unfortunately, without providing detailed information how this arbitrary threshold was selected. Beside further information needed and without knowing the rationale for an optimal threshold, it is suggested to split the entire cohort into quartiles which might reflect prediction power closer to what has been shown in the manuscript. How does a quartile split look like?

As referenced above (Reviewer #1, Q3), we have now included univariate and multivariate Cox proportional hazard models to assess the association between plasma Cav-1-sphingolipid signature scores with progression free survival in AS subjects. Age, 5-alpha reductase treatment, and baseline tumor volume, were included as co-variables in multivariate Cox proportional hazard models. Co-variables included in the model were selected using a backward stepwise method (likelihood ratio). Results are provided in **Supplementary Table S3**.

In order to find the cut-off point for the covariate that gives the largest difference between individuals in the two already defined groups, we used the method that has been described in Contal and O’Quigley (1). Using log rank statistic-based on the groups defined by cut-off we have:

$$S_k = \sum_{i=1}^D [d_i^+ - d_i \frac{r_i^+}{r_i}]$$

Where D is the total number of distinct events (disease progression (DP)), d_i is the total number of DP at each event time (t_i), d_i^+ is the total number of DP when the Cav-1-sphingolipid signature value is bigger than the cut-off point. r_i and r_i^+ also define as the total number at risk for all Cav-1-sphingolipid signature values and Cav-1-sphingolipid signature values larger than cut-off point respectively.

We calculated S_k for all possible cut point in Cav-1-sphingolipid signature column and the estimated cut point is the value that yields the maximum S_k . In our analysis, the maximum value of S_k is at the top 16.4% of Cav-1-sphingolipid signature values. In another word, top 16.4% of Cav-1-sphingolipid signature values are in the high-risk group and other 83.6% are in the low-risk group.

In order to calculate the p -value of this test we used the following formula and it gives the value of 0.009. It suggests that the Cav-1-sphingolipid signature level highly relates to progression free survival.

$p\text{-value} \approx 2\exp(-2Q^2)$
 where:

$$Q = \frac{\max |S_k|}{s\sqrt{D-1}}$$

and

$$s^2 = \frac{1}{D-1} \sum_{i=1}^D \left\{ 1 - \sum_{j=1}^i \frac{1}{D-j+1} \right\}^2$$

We have added this information to the main text as well as the supplementary material.

3. The authors have accessed the TCGA prostate cancer transcriptomic data to provide further evidence of CAV-1 in prostate cancer and to study the relationship to lipid metabolism. Here, clinicopathological data accompanied with the data sets should be thoroughly investigated in regards to CAV-1/ gene sets. Has this been done? Neither the results nor the figure legends provide details on the generation of “iCluster-1 to 3”. An appropriate dendrogram on the patients and possible prostate cancer subgrouping (if there are any) might be of benefit.

We would like to provide clarify regarding the intended use of TCGA prostate cancer transcriptomic data and the iClusters. The primary intent of TCGA prostate cancer transcriptomic data was to interrogate putative relationships between Cav-1 and gene expression profiles reflective of lipid managing apparti. We found that high CAV1 mRNA expression was found to be positively associated with gene ontologies related to lipid scavenging and metabolism, glycosylceramide metabolic process as well as the ceramide pathway (**Figure 1C**). We additionally overlaid this information with iClusters that were previously described by the Cancer Genome Atlas Network to reflect molecular heterogeneity that exists among primary prostate cancers¹⁴. Our results indicated that high CAV1 mRNA expression was predominately associated with iCluster 3, which is characterized by elevated PI3K/AKT, MAP-Kinase and receptor tyrosine kinase activity¹⁴. Our group, and others, have previously demonstrated the role of Cav-1 and tumor progression^{2, 4, 15, 16, 17}.

We have included this information in the main text of the manuscript.

4. The Western blot Figure 2B (bottom of the images) has to be improved. Here, at least a repetitive Western blot with reference protein should be shown for all three cell lines together. In addition, an appropriate quantification should be done on the images. Stating “...exhibited substantially higher fluorescence accumulation” is insufficient. In addition, the scale bar in cell culture images is missing.

We note that immunoblots for baseline Cav-1 in RM-9 and PC-3M are included in Figure 2A. Immunoblots for baseline Cav-1 in LNCaP is provided in Supplementary Figure S3A.

5. There is no clear evidence that neither overexpression or silencing of Cav-1 in selected cell lines is functional. The Cav-1 antibody seems to work well in Western blot (Figure 2A). Thus, the overexpression and silencing could be easily demonstrated in all cell lines used throughout this study.

We have added immunoblots depicting total cell lysate levels of Cav-1 following overexpression or silencing of Cav-1 in LNCaP and PC-3M prostate cells, respectively. These are now provided in **Supplementary Figure S3A**.

6. CAV1-dependent investigation of the CCLE and TCGA in Figure 2 has been done in order to study the possible association of CAV1 and metabolite enzyme-encoding gene expression. Here, the description in the text (row 189 to 196) is not obvious in the related figures. It might be true for glycosphingolipids, but not for ceramides. A more sophisticated analysis should be done. Moreover, the stratification into high or low CAV1 is insufficiently explained. Why were only the highest and lowest quartiles chosen? What are the remaining patient numbers in both groups? Is this statistically tested?

For TCGA, we chose to use quartiles in comparison of the bottom 25th and top 25th percentiles of CAV1 expression as to highlight the effect between the most differential populations. We acknowledge that this representation only highlights a subset of the data. To address this, we have now additionally included spearman correlation analyses (**Supplementary Table S6**) based on the entire TCGA prostate cancer dataset using continuous values for CAV1 mRNA expression and mRNA expression of enzymes depicted in Figure 2E (now **Figure 3B**).

For CCLE data, given the limited number of cell lines (n=8), we stratified cell lines based on mean CAV1 mRNA expression into those with very high mRNA expression (log₂ values >11 (range 11.01-13.61)) versus those with very low mRNA expression ((log₂ values <7 (range 4.16-6.88)). We have clarified this point in the revised manuscript.

Based on these analyses, we found that mRNA expression of enzymes involved in the biosynthesis of glycosphingolipids, including glucosylceramide synthase (UGCG), and lactosylceramide synthases B4GALT5 and B4GALT6, were elevated in Cav-1-high prostate

cancer cell lines and prostate tumors. These findings collectively favor the notion that Cav-1 high prostate cancer cell lines and prostate tumors exhibit elevated levels of glycosphingolipids, important constituents of our sphingolipid signature (**Figure 1A-B**). Interestingly, as the Reviewer points out and as we describe in the main text, gene expression of enzymes that facilitate biogenesis of the precursor, ceramide, tended to be lower in CAV1 high prostate cancer cell lines and prostate tumors. Despite reductions in mRNA levels of sphingomyelinases, our prior analysis between gene expression of CAV1 and ontologies related lipid metabolism indicated positive associations with lipid scavenging (**Figure 1C**). Similarly, we demonstrate that Cav-1 high prostate cell lines PC-3M and RM-9 exhibit substantially higher uptake of fluorescent 1,1'-dioctadecyl-3,3,3',3'-tetramethylindocarbocyanine (DiI)-conjugated synthetic self-assembled lipid particles (SSALPs) in comparison to Cav-1 low LNCaP (**Figure 2B**). Therefore, we interrogated whether uptake of extracellular sphingomyelins provides an appreciable source of ceramides and their glycosphingolipid derivatives through the use of sphingomyelin(d18:1/18:1)-deuterium(d)9 enriched SSALPs; the results of which confirmed that extracellular sphingomyelins are indeed a source of ceramides and their glycosylated derivatives (**Figure 2D**).

7. The authors utilize LC-MS for tracing uptake of extracellular sphingomyelins and subsequent synthesis of ceramides and glycosphingolipids in the three prostate cancer cell lines. Drawing conclusions from the cell lines based on their endogenous Cav-1 expression doesn't seem to be appropriate. Thus, the authors should show in at least 2 cell lines that their results are Cav-1 (in-)dependent by either silencing or overexpressing Cav-1. In general, comparing cell lines based on their Cav-1 expression only, does not adequately allow to address the function of Cav-1 as differences might be unrelated to the protein. Here, additional functional tests as already performed at the beginning should be consequently applied in almost all the experiments.

We acknowledge the Reviewer's critique as valid. We do not imply that Cav-1 is altering the biochemical fate of sphingomyelin. Instead, we demonstrate that extracellular sphingomyelins are an appreciable source of ceramides and their glycosylated derivatives. Based on our in vitro experiments, we provide experimental evidence that Cav-1 participates in scavenging of extracellular sphingomyelin (**Figure 4B**) and secretion of Cav-1-sphingolipid enriched extracellular vesicles. As per the Reviewer's recommendation, we assessed whether knockdown of CAV1 in PC-3M would alter the rate by which sphingomyelin(d18:1/18:1)-deuterium(d)9 is catabolized to ceramide(d18:1/18:1)-d9 and subsequently glycosylated to glucosylceramide(d18:1/18:1)-d9. No statistically significant differences were observed in the levels of ceramide(d18:1/18:1)-d9 following knockdown of CAV1 in PC-3M prostate cells, although glucosylceramide(d18:1/18:1)-d9 tended to be lower. This implicates that the catabolism of sphingomyelin to ceramide is independent of Cav-1.

8. For entire figure 3, scale bar is missing and statistical evaluation is needed. The number of experiments is neither stated in the results nor figure legend.

We have now added intensity scales and size scale bars (50 μ m) to the micrographs in figure 3 (now **Figure 4**). Experiments were conducted at least in duplicate. The figure presents representative fluorescent images captured via confocal and widefield microscopy using live cell imaging techniques. Panels within the figure compare lipid uptake (B), lysosome and mitochondria architecture (C), and ROS accumulation (D) and comprise images obtained during the same respective imaging session from the same multi-well imaging plate. Each panel presents images of cells that were seeded and treated with siRNAs at the same time; imaging reagents were added the same uniform time point and analyses were all performed at the same imaging session. Digital image acquisition parameters and look up table mappings were uniformly set for all images within each respective panel. In panels A and C, we highlight individual cells of interest, but also include large fields of view to demonstrate the observations are conserved among replicate cells under each given treatment or culture condition.

9. “Selective” inhibitors of glucosylceramide synthase (UGCG) were applied to test a targetable metabolic vulnerability. The literature on the specificity of UGCG inhibitors is not straightforward, especially in regards to the two inhibitors used. However, it is quite evident that inhibition of UGCG does not reduce all glucosylceramide-related glycosphingolipids. Also, studies focused on UGCG are quite controversial. The authors found cytotoxicity with accompanied alteration of metabolites in 2 cell lines. Again, other non-glycosphingolipid related drugs might cause the same effect. Have other drugs been tested with focusing on identical readout? Are PPMP and PDPM working the same way as shown for eliglustat?

We agree with the Reviewer that the ‘selectivity’ of UGCG inhibitors is not straightforward and we are aware that caution should be used when interpreting findings¹⁸. To this end, we intentionally tested 3 different commonly used UGCG inhibitors (PPMP, PDMP and Eliglustat) as a means to attenuate potential off-target effects.

Herein, we demonstrate that challenge of RM-9 and PC-3M prostate cells with PPMP, PDMP or Eliglustat induced acute statistically significant elevations in intracellular ceramides whereas global pools of glycosphingolipids were reduced, consistent with UGCG inhibition (**Figure 6C top and bottom panels**). The Reviewer is accurate that not all glucosylceramide-related glycosphingolipids were reduced following UGCG inhibition; however, we note that lipidomic analyses were conducted following 6hr challenge of RM-9 and PC-3M prostate cancer cells with the respective UGCG inhibitors. We intentionally evaluated the short term effects of UGCG inhibition to capture early metabolic alterations and as to avoid secondary events that may cause elevations in ceramide pools^{19, 20, 21}, and to prevent reductions in GLS expression which can occur as a result of reduced cell survival¹⁸. It is anticipated that this short time frame would likely not translate into reductions in all glucosylceramide-derived glycosphingolipids. Regardless, global abundances of glycosphingolipids were, in general, lower following short-term treatment of prostate cancer cell lines with UGCG inhibitors.

The Reviewer is correct that other drugs may similarly result in increased ceramide pools, particularly given the relationship between ceramide-species and apoptosis^{19, 20, 21}. As referenced above, we therefore intentionally evaluated the short term effects of UGCG inhibition to capture acute metabolic alterations.

In relation to whether or not PPMP and PDMP are working in the same way as eliglustat, based on our lipidomic analyses, all three drugs elicited rapid increases in ceramide pools whereas glycosphingolipids in general were reduced suggesting a uniformity in response. However, it is also evident based on the viability curves and lipidomic results for RM-9 and PC-3M cells following treatment with PPMP, PDMP or eliglustat, that PPMP and PDMP are more efficacious than eliglustat on the observed phenotype when considering equivalent concentrations. This is likely attributed to the fact that PDMP and PPMP are structural analogs of ceramide and may therefore impact other enzymes involved in glycosphingolipid metabolism, or may directly function similar to endogenous ceramides²².

We have now commented upon these considerations in the main text of the revised manuscript.

Does Cav-1 overexpression (e.g. LNCaP) result in enhanced resistance to eliglustat?

We have now added viability (MTS assay) of LNCaP cells treated with eliglustat following overexpression of Cav-1. Overexpression of Cav-1 resulted in reduced the anti-cancer efficacy of eliglustat (**Figure 6G**).

10. It is not clear where “... and acute (6hr) accumulation of intracellular ceramides...” (statement row 282-285) is demonstrated. There are no data on detailed expression of metabolites as stated.

Intracellular levels of ceramides are provided in **Figure 5A-C (now figure 6C; top panel)**. We have now included a supplemental table (**Supplementary Table S7**) that provides area abundances (mean +/- stdev), fold-changes and individual p-values of individual lipid species following 6hr treatment of PC-3M and RM-9 prostate cancer cells with either vehicle, eliglustat, PDMP or PPMP.

Like eliglustat, PPMP and PDMP similarly induced acute elevations in intracellular ceramides (**provided in upper panel of Figure 6C of revised manuscript; Supplementary Table S7**) and reductions in intracellular glycosphingolipid pools (**provided in lower panel of Figure 6C of revised manuscript**).

11. The colors for treatment (eligustat, PDMP, PDMP, vehicle) in figure 5B should be identical.

We have fixed figure 5b (now **Figure 6B**) to ensure consistency in the colors.

11. Different series of glycosphingolipids play various roles in cancer. The limited number of glycosphingolipids investigated does not allow drawing conclusions for glycosphingolipids in general taken the heterogeneity of this class of biomolecules. Thus, it would be of benefit to reflect findings reported with the literature on glycosphingolipids in the context of cancer in the discussion.

The Reviewer is correct that there exists considerable heterogeneity and complexity in discrete glycosphingolipid species. We have added additional consideration for this important topic in our discussion. In future studies we aim to more fully explicate the cancer-supportive features of the onco-metabolic network we report here, including glycosphingolipid complexity and functional roles.

Reviewer #3 (Remarks to the Author):

In this study, Vykoukal et al identify a Cav-1 and sphingolipid/glycosphingolipids and sphingomyelin) that is prognostic of prostate cancer disease progression and present a potentially interesting mechanism whereby prostate cancer cells utilize Cav-1 and increased ceramide to glycosphingo conversion to mitigate mito-toxicity potentially via increased clearance of ceramides. In addition they replicate the plasma signature of patients in tumor bearing mice.

I found the data to be quite interesting. The paper covers a lot of ground and proposes a fairly detailed mechanism. Various technical points need to be addressed, and the mechanism could be refined a bit more. Altogether this is a good mixture of clinical and basic studies. Comments are on the heavy side, but this has as much to do with our interest as it does with science.

Major comments:

1. I am somewhat confused by their overarching mechanism: The authors demonstrate nicely that tumor cells upregulate ceramide synthesis from SM, but cells must then deal with (ceramide induced) mitochondrial toxicity by shuttling out glycosphingolipids. The mechanism (as written) doesn't make sense to me, as it is a futile cycle with large bioenergetic/biosynthetic costs. If the ceramide toxicity point is important to the authors, they should test this with rescue studies (perhaps with inhibitors?). There are various other reasons why mitochondria are damaged, so I think they can work around it. Along these lines, are mitophagy markers generally upregulated in Cav1 KD cells?

We appreciate this insightful Reviewer comment and drawing attention to the fact that our original discussion of the overarching mechanism was somewhat indefinite with regard to ceramide shunting into glycosphingolipids. We now clarify that sphingomyelin – ceramide – glycosphingolipid pathway functions to support biogenesis of Cav-1-sphingolipid-enriched vesicles (Cav-LPs).

Our mechanistic model posits a role for Cav-1 in mitochondrial quality control that includes clearance of damaged mitochondria components through secretion of Cav-1-sphingolipid particles. This is consistent with an emerging view of mitochondrial maintenance as comprising different scales of quality control that include bulk elimination of mitochondria as well as targeted elimination of damaged parts of the mitochondrial network or specific mitochondrial proteins³. Such selective editing of mitochondria would provide an efficient response to oncogenic stress and means to evade lethal mitochondrial dysfunction. PINK1, Parkin and LC3 have been found to participate in formation of mitochondrial-derived vesicles that enable quality control at the sub-organelle level by allowing mitochondria to actively shed damaged fragments in addition to known roles in targeting damaged mitochondria for autophagic elimination^{22, 23, 24, 25, 26}.

It is becoming increasingly evident that multi-tiered programs of mitochondrial quality control operate at the cell, organelle, sub-organelle and protein levels and that these programs intersect with other quality control pathways including the unfolded protein response, shedding of vesicles, proteolysis, and degradation by the ubiquitin–proteasome system³.

We now have supplemented our model (now **Figure 8**) by designating modules to delineate specific processes that contribute to the overall Cav-1 mediated onco-metabolic cycle we present. Module 1 describes a role for Cav-1 in mediating extracellular lipid influx. Module 2 encompasses increased tumor catabolism of extracellular sphingomyelins and altered ceramide metabolism that result in an increased glycosphingolipid synthesis that we observe in cancer cells with high intrinsic Cav-1 levels. Module 3 describes efflux of Cav-1-sphingolipid particles that are enriched in mitochondria-associated proteins and lipid metabolites. Within each module are multiple, interdependent, and potentially rate-limiting biochemical steps that are required for the metabolic progression pathway we describe in our manuscript. Throughout the years, focused research on the role of Cav-1 in cancer cell metabolism and the effects of these metabolic activities on the tumor microenvironment have been encumbered by the widespread functions of Cav-1. We submit that the results of our study have deconvoluted the role of Cav-1 in prostate cancer lipid uptake and metabolism and revealed novel therapeutic vulnerabilities within each of the modules we describe. We hope the Reviewers agree that including a schema (**Figure 8**) that delineates and interrelates Cav-1 regulated lipid metabolic functions facilitates a better understanding of the novel results of our study and the overall role of Cav-1 in prostate cancer lipid metabolism.

2. Figure 2G must include the abundance of total (labeled and unlabeled) species to make any claim about fluxes. The LnCAPs could have a much larger pool size (though I think results will hold). The 10-fold abundance claim about d9-ceramide is misleading without this information.

We have added the abundance of the labeled and unlabeled species. We note that naturally occurring sphingomyelin(d18:1/18:1) and ceramide(d18:1/18:1) was relatively low in comparison to the -d9 isotopologues. We have additionally included the ratio between the peak area of ceramide-d9 relative to the peak area of sphingomyelin-d9. The ratio of ceramide(18:1/18:1)-d9 to sphingomyelin(18:1/18:1)-d9, based on peak area, was appreciably higher in PC-3M (ratio: 0.14) and RM-9 (ratio: 0.23) as compared to LNCaP (ratio: 0.02), demonstrating an overall higher metabolic flux into ceramide biosynthesis via sphingomyelin salvaging (**Figure 2F**).

3. This important and elegant experiment would be better tested in isogenic Cav1 +/- cells (as in Fig 2C/D). As currently shown, the Cav1 expression is correlative with their flux findings. These 3 cell lines are as different as 3 PCa patients – different genetic backgrounds, etc.

We acknowledge the Reviewer's critique as valid. We do not imply that Cav-1 is altering the biochemical fate of sphingomyelin. Instead, we only demonstrate that extracellular sphingomyelins are an appreciable source of ceramides and their glycosylated derivatives. Based on our in vitro experiments, we demonstrate that Cav-1 mediates uptake of extracellular sphingomyelin (**Figure 4B**). This insinuates that Cav-1 mediates substrate (i.e. sphingomyelin) bioavailability. To this end, as per the Reviewer's recommendation, we assessed whether knockdown of *CAV1* in PC-3M would alter the rate by which sphingomyelin(d18:1/18:1)-deuterium(d)9 is catabolized to ceramide(d18:1/18:1)-d9. There was no statistically significant differences were observed in the

levels of ceramide(d18:1/18:1)-d9 following knockdown of *CAV1* in PC-3M prostate cells, although glucosylceramide(d18:1/18:1)-d9 tended to be marginally lower. This implicates that the catabolism of sphingomyelin to ceramide is independent of Cav-1.

4. The impact of Cav1 modulation on cell growth should be shown. This also puts the flux and metabolomics data in better context. Is this dependent on lipids in the medium? It would provide the reader with more confidence in their mechanism and data in Figure 3. I like the mitochondria efflux concept, but this is not supported enough by the data provided.

The addition of lipid particles to serum-free media does promote increased proliferation of PC-3M and RM-9 prostate cancer cells. Similarly, we demonstrate that extracellular lipid bioavailability modulates Cav-1 protein expression (**Figure 2A**).

Knockdown or overexpression does not impact overall proliferation over the course of 48 hours. On the basis of relatively robust cell growth in serum-free medium we speculate that RM-9 and PC-3M cells secrete survival factors that maintain a stable growth rate during the 48 hour assay period—overcoming the effects genetically reduced or increased Cav-1 levels.

5. Are significant amounts of mitochondrial DNA or protein present in EVs? Not +/- Cav1 (this will certainly be the case), instead the authors should compare vesicular mitochondrial pools to total to determine whether this flux is appreciable.

The Reviewer raises an interesting point regarding presence of mitochondrial DNA in EVs. The potential cancer biomarker utility of this specific form of cell free DNA has recently been explored by a number of groups. The Reviewer also suggests a more comprehensive accounting of mitochondria-derived EV protein composition and flux relative to total circulating EVs. We agree these are interesting areas of intersection with the studies we report here and we will indeed pursue these experiments in the course of our future studies; however, these remain outside of the scope of the current investigation.

6. If Cav-1 protects against mito toxicity, is overall mito function altered with Cav-1 KD such as oxygen consumption etc.?

The Reviewer raises a valid question regarding mitochondrial function in the context of Cav-1 expression. Our model includes a role for Cav-1 in supporting mitochondrial dynamics and we do employ live cell imaging studies of Cav-1 knockdown PC-3M cells using mitochondrial imaging probes to demonstrate accumulation of punctate mitochondria 27, 28, 29 and a reduction in prevalence of lysosomes (Figure 4C). Using ROS-sensitive probes, we further demonstrate that these changes were accompanied by an increase in intracellular reactive oxygen species (Figure 4D). Additional, focused experiments to specifically interrogate the relationship between mitochondrial function and metabolic flux and Cav-1 expression are indeed warranted, but, in our opinion, outside of the scope of the current report.

Nevertheless, we note that previous quantitative computational imaging studies by others have established a well-defined inverse relationship between OCR/ECAR rates and the prevalence of punctate mitochondria²⁷. Also, punctate morphology was found to correlate with increased glycolysis levels and decreased oxygen consumption, suggesting mitochondrial morphology as a robust indicator of mitochondria fitness and metabolic capacity.

7. The clinical data is impactful, but none of the metabolites in question were statistically significant in their initial analysis (as the authors note). I am fine with the subsequent grouping/signature that focuses on ceramide metabolism, but these trends should be considered in the context of other clinical parameters. Could obesity or BMI be a driver of the dyslipidemia? Most plasma lipids are on liver-derived vLDL particles and thus reflects hepatic metabolism more than the tumor. On that front, it would be helpful to quantify the amount of sphingolipids in patient EVs vs vLDLs to know what they are observing in the clinical samples.

The author raises a valid concern in regards to the influence of obesity or BMI on dyslipidemia. To this end, we note that, in our AS cohort, BMI was not associated with disease progression (DP (HR: 1.02, 95% CI: 0.40-2.64, *p*-value: 0.965) (**Supplementary Table S3B**) indicating that our lipid signature is unlikely to be biased by obesity alone. We also agree that detailed characterization of subpopulations of lipoprotein particles and EVs would be informative. We have now added additional data presenting lipidomic analyses of prostate cancer cell line derived EVs cultured under lipoprotein-depleted condition (**Figure 5F**). These analyses demonstrate sphingolipids to be highly enriched in cancer cell secreted EVs. With regards to the specific question of VLDLs vs EVs, it is likely that differential distribution of specific lipids and lipid classes exists among circulating lipid particles and would be biologically relevant. However, enumeration and characterization of these particle subclasses presents a distinct technical challenge and is an ongoing area of research that is currently being developed. As these approaches mature, we will be better able to answer this highly relevant question.

Minor points:

1. It's not my area (making it all the more important), but the FL image results weren't clear to me initially. I understand the point from the text and see the image, but it could be presented better. Some quantitation of FL imaging would allow the authors to show the highlight the difference to the reader more clearly.

We have revised the fluorescent images in **Figures 2** and **4** to more clearly present our findings. To facilitate quantitative evaluation of the fluorescent images, we now include additional intensity scales and size scale bars to the micrographs. We have also assembled the panels at higher resolution based on the original native confocal and widefield fluorescent digital captures.

2. How was the lipid abundance in figure s3, 2c-d, 2g normalized, was this to cell number or protein?

Lipid abundances were normalized based on total cell number. We recognized that differences in cell size can bias normalization by absolute cell number. We note that there was no appreciable difference in the median cell diameter of PC-3M and LNCaP cells (median diameter 19.53 vs 19.96, respectively), thereby reducing such bias.

3. Authors note “Cav-1-mediated clearance of ‘damaged’ mitochondria”, then would you expect less LC3BII with eliglustat?

As an inhibitor of glucosylceramide synthase (UGCG), eliglustat impedes ceramide to glycosphingolipid conversion. Metabolomic profiling of RM-9 and PC-3M cell lines following treatment with eliglustat (Figure 6C) indicates decreased abundance of intracellular glycosphingolipids along with a concomitant increase in intracellular ceramides. Ceramide has been shown to be necessary and sufficient to induce lethal autophagy by anchoring LC3B-II-autophagolysosomes to mitochondrial membranes²⁸. Induction of ceramide accumulation and increased intracellular LC3B-II following eliglustat treatment are consistent with this mechanistic shift from survival to lethal mitophagy with anti-tumor effect.

4. Please add a scale bar to 6f images

Scale bar has been added to figure 6f (now **Figure 7F**).

5. Please plot individual values on 6 b-c so that the biological variation is clear.

We have plot individual values for Figure 6 B-C (now **Figure 7B-C**) as to demonstrate biological variation. With regard to experimental variation in IVIS data, on the basis of our experience with this and other similar model systems we expected this variation due to vascular heterogeneity and regionally variable necrosis—due to rapid growth, combined with luciferin penetration into the tumor.

References

1. Baley PA, Yoshida K, Qian W, Sehgal I, Thompson TC. Progression to androgen insensitivity in a novel in vitro mouse model for prostate cancer. *The Journal of steroid biochemistry and molecular biology* **52**, 403-413 (1995).
2. Watanabe M, *et al.* Functional analysis of secreted caveolin-1 in mouse models of prostate cancer progression. *Molecular cancer research : MCR* **7**, 1446-1455 (2009).
3. Pickles S, Vigie P, Youle RJ. Mitophagy and Quality Control Mechanisms in Mitochondrial Maintenance. *Current biology : CB* **28**, R170-r185 (2018).
4. Yang G, Truong LD, Wheeler TM, Thompson TC. Caveolin-1 expression in clinically confined human prostate cancer: a novel prognostic marker. *Cancer research* **59**, 5719-5723 (1999).
5. Karam JA, Lotan Y, Roehrborn CG, Ashfaq R, Karakiewicz PI, Shariat SF. Caveolin-1 overexpression is associated with aggressive prostate cancer recurrence. *The Prostate* **67**, 614-622 (2007).
6. Liu JM, Cheng SH, Liu XX, Xia C, Wang WW, Ma XL. Prognostic value of caveolin-1 in genitourinary cancer: a meta-analysis. *International journal of clinical and experimental medicine* **8**, 20760-20768 (2015).
7. Wang T, *et al.* JAK/STAT3-Regulated Fatty Acid beta-Oxidation Is Critical for Breast Cancer Stem Cell Self-Renewal and Chemoresistance. *Cell metabolism* **27**, 136-150.e135 (2018).
8. Aicheler F, Li J, Hoene M, Lehmann R, Xu G, Kohlbacher O. Retention Time Prediction Improves Identification in Nontargeted Lipidomics Approaches. *Analytical chemistry* **87**, 7698-7704 (2015).
9. Musunuru HB, *et al.* Active Surveillance for Intermediate Risk Prostate Cancer: Survival Outcomes in the Sunnybrook Experience. *The Journal of urology* **196**, 1651-1658 (2016).
10. Peterschmitt MJ, Freisens S, Underhill LH, Foster MC, Lewis G, Gaemers SJM. Long-term adverse event profile from four completed trials of oral eliglustat in adults with Gaucher disease type 1. *Orphanet journal of rare diseases* **14**, 128 (2019).
11. Bartz R, Zhou J, Hsieh JT, Ying Y, Li W, Liu P. Caveolin-1 secreting LNCaP cells induce tumor growth of caveolin-1 negative LNCaP cells in vivo. *International journal of cancer* **122**, 520-525 (2008).
12. Tahir SA, *et al.* Secreted caveolin-1 stimulates cell survival/clonal growth and contributes to metastasis in androgen-insensitive prostate cancer. *Cancer research* **61**, 3882-3885 (2001).

13. Lin CJ, *et al.* The paracrine induction of prostate cancer progression by caveolin-1. *Cell death & disease* **10**, 834 (2019).
14. The Molecular Taxonomy of Primary Prostate Cancer. *Cell* **163**, 1011-1025 (2015).
15. Tahir SA, *et al.* Preoperative serum caveolin-1 as a prognostic marker for recurrence in a radical prostatectomy cohort. *Clinical cancer research : an official journal of the American Association for Cancer Research* **12**, 4872-4875 (2006).
16. Goetz JG, *et al.* Biomechanical remodeling of the microenvironment by stromal caveolin-1 favors tumor invasion and metastasis. *Cell* **146**, 148-163 (2011).
17. Gumulec J, *et al.* Caveolin-1 as a potential high-risk prostate cancer biomarker. *Oncology reports* **27**, 831-841 (2012).
18. Alam S, Fedier A, Kohler RS, Jacob F. Glucosylceramide synthase inhibitors differentially affect expression of glycosphingolipids. *Glycobiology* **25**, 351-356 (2015).
19. Yabu T, *et al.* Stress-induced ceramide generation and apoptosis via the phosphorylation and activation of nSMase1 by JNK signaling. *Cell death and differentiation* **22**, 258-273 (2015).
20. Ghafourifar P, *et al.* Ceramide induces cytochrome c release from isolated mitochondria. Importance of mitochondrial redox state. *The Journal of biological chemistry* **274**, 6080-6084 (1999).
21. Snider JM, *et al.* Multiple actions of doxorubicin on the sphingolipid network revealed by flux analysis. *Journal of lipid research* **60**, 819-831 (2019).
22. Shayman JA, Abe A, Hiraoka M. A turn in the road: How studies on the pharmacology of glucosylceramide synthase inhibitors led to the identification of a lysosomal phospholipase A2 with ceramide transacylase activity. *Glycoconjugate journal* **20**, 25-32 (2004).
23. Burman JL, *et al.* Mitochondrial fission facilitates the selective mitophagy of protein aggregates. *The Journal of cell biology* **216**, 3231-3247 (2017).
24. Yamashita SI, *et al.* Mitochondrial division occurs concurrently with autophagosome formation but independently of Drp1 during mitophagy. *The Journal of cell biology* **215**, 649-665 (2016).
25. Soubannier V, *et al.* A vesicular transport pathway shuttles cargo from mitochondria to lysosomes. *Current biology : CB* **22**, 135-141 (2012).

26. McLelland GL, Soubannier V, Chen CX, McBride HM, Fon EA. Parkin and PINK1 function in a vesicular trafficking pathway regulating mitochondrial quality control. *The EMBO journal* **33**, 282-295 (2014).
27. Giedt RJ, *et al.* Computational imaging reveals mitochondrial morphology as a biomarker of cancer phenotype and drug response. *Scientific reports* **6**, 32985 (2016).
28. Sentelle RD, *et al.* Ceramide targets autophagosomes to mitochondria and induces lethal mitophagy. *Nature chemical biology* **8**, 831-838 (2012).

Reviewers' comments:

Reviewer #1 (Remarks to the Author):

The revisions to the manuscript have enhanced the clarity and quality of the manuscript text and key figures. The critical amendments I requested to the clinical descriptors are now accurate and there is greater detail about the key experimental and analysis details that were previously missing or incomplete. The inclusion of quantitative data for the animal studies is appreciated and markedly strengthens this aspect, but a major issue raised by myself and another reviewer remains unaddressed- the quantification of data and confirmation of CAV1 modulation in the immunofluorescent studies in Figure 4. With relatively few cells being shown in the images, it is particularly important to include quantitative data with larger numbers of cells- this is now relatively routine for immunofluorescence work thus it is surprising it has not been done to support a major conclusion of the manuscript. Regarding CAV-1 detection, despite the western blots now included for an independent experiment, given the known heterogeneity of expression on an individual cell level in even stably transfected lines, CAV-1 levels in the images for Figure 4 remain an important experiment to include to ensure the data and changes are interpreted appropriately. For that purpose live cell imaging is not necessary and standard immunofluorescent protocols could be used to visualize CAV1 and the other markers included.

Other Comments:

1. The authors have answered the queries regarding the metabolomics analysis of the clinical samples used in the study. Some of these issues (patient fasting status, freeze-thawing etc) are intrinsic to large clinical cohorts of active surveillance and understandably cannot be revisited by the investigators. However for benefit of the readers and interpretation of results compared to other cohorts, it would be important to include these details in the materials and methods section of the manuscript, not only the rebuttal.

2. Figure 1C is now much clearer. The data for the normalized gene expression should be included as supplementary, and the authors could consider expressing the heatmaps data as median rather than average values, as the average will be much more strongly influenced by outlier or high abundance genes (or both).

Minor comments:

1. For Figure 5B (now Figure 6B), despite some improved labeling, the figure is proportionally far too small compared to the other figure panels and is again impossible to read and interpret.

2. Line 350: Figure 5E should be Figure 6E.

Reviewer #2 (Remarks to the Author):

The authors of the revised manuscript entitled “Cav-1-Mediated Sphingolipid Oncometabolism Underlies a Metabolic Vulnerability of Prostate Cancer” have appropriately addressed all questions raised by the reviewer.

Reviewer #3 (Remarks to the Author):

The authors have provided a fairly thorough and verbose (longer is not always better) response to our critiques, arguing away most points. See some minor issues below. I would argue that “mitochondria” is a central part of their mechanism (I see the term repeated lots in Figure 7), why is a simple validation of actual mitochondrial components in their EVs beyond the scope of this paper?

They propose an interesting and complex mechanism. The link to mitochondria isn't so strong and probably not critical here, so I would suggest backing off some of those aspects. It's not just mitochondria that need turnover. While I think questions remain about some specific details, in my view the paper and concept should be published, but their claims need to be kept in check. I am specifically referring to the abstract sentence:

“Our findings demonstrate involvement of this metabolic program in support of vesicle formation that facilitates trafficking of ‘damaged’ mitochondrial components from the cell.”

This has not been shown in my view, although they hope this is the mechanism. I would cut or show me some “damaged” mitochondrial components and their functional impact.

Some minor corrections:

There is a typo (line 350), where we think Figure 5E should be 6E

For figures 2D/F, using the term “relative abundance” for the axis title would be much better, as “area units” is vague. Please explain how these were normalized in the legend for the reader.

Reviewer #1 (Remarks to the Author):

Concerns: The revisions to the manuscript have enhanced the clarity and quality of the manuscript text and key figures. The critical amendments I requested to the clinical descriptors are now accurate and there is greater detail about the key experimental and analysis details that were previously missing or incomplete. The inclusion of quantitative data for the animal studies is appreciated and markedly strengthens this aspect, but a major issue raised by myself and another reviewer remains unaddressed- the quantification of data and confirmation of CAV1 modulation in the immunofluorescent studies in Figure 4. With relatively few cells being shown in the images, it is particularly important to include quantitative data with larger numbers of cells- this is now relatively routine for immunofluorescence work thus it is surprising it has not been done to support a major conclusion of the manuscript. Regarding CAV-1 detection, despite the western blots now included for an independent experiment, given the known heterogeneity of expression on an individual cell level in even stably transfected lines, CAV-1 levels in the images for Figure 4 remain an important experiment to include to ensure the data and changes are interpreted appropriately. For that purpose live cell imaging is not necessary and standard immunofluorescent protocols could be used to visualize CAV1 and the other markers included.

Response: We now provide quantitative data for confocal images depicted in **Figure 4** in **Supplementary Figure S5A, C-D**. We performed image analysis with NIS Elements software. The entire z-stack was subtracted for background and a threshold was defined based on intensity to select areas containing the signal of interest. Mean intensity was calculated for each object (from threshold areas) in all planes for each condition, and 'intensity distributions' plotted using violin plots. Statistical significance was determined using One-way ANOVA; pairwise comparisons were performed using Tukey HSD multiple comparison test and adjusted p-value reported.

We have additionally included representative co-immunofluorescence images for CAV1 (FITC) and mitochondrial potential/ROS (assessed by MitoTracker Red CMXRos [1]) following siRNA-mediated knockdown of *CAVI* in PC-3M prostate cancer cells. Consistent with our original findings, siRNA-mediated knockdown of *CAVI* reduced surface CAV1-GFP positivity, whereas mitochondrial ROS were significantly increased. We believe these findings warrant inclusion in the primary figures and we have now included this data in **Figure 4D**. Using MitoTracker Green FM, we further confirmed that the elevation in mitochondrial ROS was similarly met with buildup of 'punctate' mitochondrial mass following siRNA-mediated knockdown of *CAVI* in PC-3M prostate cells (now provided in **Supplementary Figure S5B**), consistent with our prior findings (**Figure 4C**).

Other Concerns/Comments:

1. The authors have answered the queries regarding the metabolomics analysis of the clinical samples used in the study. Some of these issues (patient fasting status, freeze-thawing etc) are intrinsic to large clinical cohorts of active surveillance and understandably cannot be revisited by the investigators. However for benefit of the readers and interpretation of results compared to other cohorts, it would be important to include these details in the materials and methods section of the manuscript, not only the rebuttal.

Response: We now include information regarding sample collection and fasting status in the primary text of the manuscript:

“EDTA was used in all plasma collections and all specimens underwent a similar number of freeze/thaw cycles prior to obtaining metabolomics data. Sample ages varied, as the study began accrual in 2006 and continued over a ten year time period. Plasma was not obtained from fasted individuals.”

2. Figure 1C is now much clearer. The data for the normalized gene expression should be included as

supplementary, and the authors could consider expressing the heatmaps data as median rather than average values, as the average will be much more strongly influenced by outlier or high abundance genes (or both).

Response: Normalized gene expression of *CAVI* and genes involved in lipid scavenging and metabolism are now provided in **Supplementary Table S6**; and expression averaged by pathway is provided in **Supplementary Table S7**. Genes were normalized by centering to standard deviation from the median across tumors.

We note that averaging by pathway versus using the median for the pathway yields the same overall result. This is because we would be working from scale-normalized logged values, where the log₂ transformation would help prevent outliers from weighing heavily on the computed average. For example, when using average vs median to define the “GO_APOLIPOPROTEIN_BINDING” total pathway score, the Pearson’s correlation between the two pathway scores has an R-value of 0.9. Individual gene values are also scaled to standard deviations from the median, and so high abundance genes would have the same relative weight as low abundance genes.

Minor comments:

1. For Figure 5B (now Figure 6B), despite some improved labeling, the figure is proportionally far too small compared to the other figure panels and is again impossible to read and interpret.

Response: Figure 6B has been enlarged to provide better clarity. We emphasize that the intent of the heatmaps is to demonstrate that treatment of RM-9 and PC-3M prostate cancer cells with UGCG inhibitors PPMP, PDMP and eliglustat induces drastic changes in the global lipidome. Individual lipid features are not easily visible in the heatmaps as several hundred annotated lipids were measured. We note that area abundances (mean +/- stdev), fold-changes and individual p-values of individual lipid species following 6 hr treatment of PC-3M and RM-9 prostate cancer cells with either vehicle, eliglustat, PDMP or PPMP are provided in **Supplementary Table S10**.

2. Line 350: Figure 5E should be Figure 6E.

Response: We thank the reviewer for recognizing this discrepancy—we have now corrected this issue.

Reviewer #2 (Remarks to the Author):

Comments: The authors of the revised manuscript entitled “Cav-1-Mediated Sphingolipid Oncometabolism Underlies a Metabolic Vulnerability of Prostate Cancer” have appropriately addressed all questions raised by the reviewer.

Response: We thank the reviewer for their positive remarks supporting publication of our findings in *Nature Communications*.

Reviewer #3 (Remarks to the Author):

Concerns/Comments #1: The authors have provided a fairly thorough and verbose (longer is not always better) response to our critiques, arguing away most points. See some minor issues below. I would argue that “mitochondria” is a central part of their mechanism (I see the term repeated lots in Figure 7), why is a simple validation of actual mitochondrial components in their EVs beyond the scope of this paper?

Response: We recognize the reviewer's comments as valid and acknowledge that we did not adequately emphasize new data that we had included in the previous revision that relates to mitochondrial components in prostate cancer cell line-derived EVs.

To determine the lipid composition and protein cargo of EVs, we performed lipidomic and proteomic analyses using mass spectrometry on EVs derived from LNCaP and PC-3M, respectively. We identified cardiolipins, important lipid constituent's exclusive to the inner mitochondrial membrane [2], in prostate cancer cell line-derived EVs (**Supplementary Table S9**). Further, evaluation of the EV-proteome identified several protein features annotated as localized to the mitochondria (**Figure 5G; Supplementary Figure S6D**). These findings provide credence that prostate cancer cells robustly secrete Cav-1-containing sphingolipid-enriched EVs [3] that are enriched in a diverse repertoire of proteins including mitochondrial-associated proteins and lipids.

Concerns/Comments #2: They propose an interesting and complex mechanism. The link to mitochondria isn't so strong and probably not critical here, so I would suggest backing off some of those aspects. It's not just mitochondria that need turnover. While I think questions remain about some specific details, in my view the paper and concept should be published, but their claims need to be kept in check. I am specifically referring to the abstract sentence:

Response: We thank the reviewer for their positive remarks and we have amended the manuscript in accordance with the reviewer's suggestions.

Concerns/Comments #3: "Our findings demonstrate involvement of this metabolic program in support of vesicle formation that facilitates trafficking of 'damaged' mitochondrial components from the cell."

This has not been shown in my view, although they hope this is the mechanism. I would cut or show me some "damaged" mitochondrial components and their functional impact.

Response: We have now removed the statement in the abstract regarding 'damaged' mitochondria. We have additionally removed 'damaged' from the main text of the manuscript as it relates to our findings. We now emphasize that the lipid oncometabolism defined in our manuscript supports increased glycosphingolipid synthesis and efflux of circulating Cav-1-sphingolipid particles containing diverse protein cargo, including mitochondrial proteins and lipids.

Some minor corrections:

1. There is a typo (line 350), where we think Figure 5E should be 6E

Response: We thank the reviewer for recognizing this discrepancy, and we have now corrected this issue.

2. For figures 2D/F, using the term "relative abundance" for the axis title would be much better, as "area units" is vague. Please explain how these were normalized in the legend for the reader.

Response: We have now revised the axis title to state 'relative abundance', and have stated how data were normalized in the figure legend. We note that for Figure 2F, the purpose was to demonstrate the metabolic fate of deuterated sphingomyelin(18:1/18:1)-d9, and not to compare differences between cell lines in relation to absolute values. Consequently, each cell line was considered its own control and the ratio of direct downstream deuterated derivatives taken to inference differences in rates of conversion.

We sincerely thank the reviewers for their insightful and important concerns and comments. Overall, the reviewers' meticulous assessment, and deep understanding of our work have resulted in substantial improvements to our manuscript.

1. Poot M, Zhang YZ, Krämer JA, *et al.* Analysis of mitochondrial morphology and function with novel fixable fluorescent stains. *The journal of histochemistry and cytochemistry : official journal of the Histochemistry Society* 1996;44(12):1363-1372.
2. Paradies G, Paradies V, De Benedictis V, *et al.* Functional role of cardiolipin in mitochondrial bioenergetics. *Biochim Biophys Acta* 2014;1837(4):408-17.
3. Skotland T, Sandvig K, Llorente A. Lipids in exosomes: Current knowledge and the way forward. *Prog Lipid Res* 2017;66:30-41.

REVIEWERS' COMMENTS:

Reviewer#1:

The authors have thoughtfully addressed the critiques. I recommend this article for publication.